# Structures of immature EIAV Gag lattices reveal a conserved role for IP6 in lentivirus assembly

**Robert A. Dick**[1�]*, **Chaoyi Xu**[2], **Dustin R. Morado**[3], **Vladyslav Kravchuk**[4], **Clifton L. Ricana**[5], **Terri D. Lyddon**[5], **Arianna M. Broad**[1], **J. Ryan Feathers**[1], **Marc C. Johnson**[5], **Volker M. Vogt**[1], **Juan R. Perilla**[2], **John A. G. Briggs**[3,6], **Florian K. M. Schur**[4,6☧]*

1 Department of Molecular Biology and Genetics, Cornell University, Ithaca, New York, United States of America, 2 Department of Chemistry and Biochemistry, University of Delaware, Newark, Delaware, United States of America, 3 Structural Studies Division, Medical Research Council Laboratory of Molecular Biology, Cambridge, United Kingdom, 4 Institute of Science and Technology Austria, Klosterneuburg, Austria, 5 Department of Molecular Microbiology and Immunology, University of Missouri, Columbia, Missouri, United States of America, 6 Structural and Computational Biology Unit, European Molecular Biology Laboratory, Heidelberg, Germany

☧ These authors contributed equally to this work.
* rad82@cornell.edu (RAD); florian.schur@ist.ac.at (FKMS)

**Data Availability Statement:** The EM-density maps and two representative tomograms have been deposited in the EMDB under accession numbers EMD-10381, EMD-10382, EMD-10383, EMD-10384, EMD-10385 and EMD-10386. The

## Abstract

Retrovirus assembly is driven by the multidomain structural protein Gag. Interactions between the capsid domains (CA) of Gag result in Gag multimerization, leading to an immature virus particle that is formed by a protein lattice based on dimeric, trimeric, and hexameric protein contacts. Among retroviruses the inter- and intra-hexamer contacts differ, especially in the N-terminal sub-domain of CA ($CA_{NTD}$). For HIV-1 the cellular molecule inositol hexakisphosphate (IP6) interacts with and stabilizes the immature hexamer, and is required for production of infectious virus particles. We have used in vitro assembly, cryo-electron tomography and subtomogram averaging, atomistic molecular dynamics simulations and mutational analyses to study the HIV-related lentivirus equine infectious anemia virus (EIAV). In particular, we sought to understand the structural conservation of the immature lentivirus lattice and the role of IP6 in EIAV assembly. Similar to HIV-1, IP6 strongly promoted in vitro assembly of EIAV Gag proteins into virus-like particles (VLPs), which took three morphologically highly distinct forms: narrow tubes, wide tubes, and spheres. Structural characterization of these VLPs to sub-4Å resolution unexpectedly showed that all three morphologies are based on an immature lattice with preserved key structural components, highlighting the structural versatility of CA to form immature assemblies. A direct comparison between EIAV and HIV revealed that both lentiviruses maintain similar immature interfaces, which are established by both conserved and non-conserved residues. In both EIAV and HIV-1, IP6 regulates immature assembly via conserved lysine residues within the $CA_{CTD}$ and SP. Lastly, we demonstrate that IP6 stimulates in vitro assembly of immature particles of several other retroviruses in the lentivirus genus, suggesting a conserved role for IP6 in lentiviral assembly.

refined models were deposited in the PDB under accession codes PDB 6T61, PDB 6T63 and PDB 6T64.

**Funding:** This work was supported by National Institutes of Health (NIH, https://www.nih.gov/) grant R01-GM107013 and National Science Foundation (NSF, https://www.nsf.gov/) grant 1659534 to V.M.V., National Institute of Allergy and Infectious Diseases (NIAID, https://www.niaid.nih.gov/) grant R01-AI147890 to R.A.D., National Institute of General Medical Sciences (NIGMS, https://www.nigms.nih.gov/) grant P30-GM110758 and National Institute of Allergy and Infectious Diseases (NIAID, https://www.niaid.nih.gov/) grant P50AI150481 to J.R.P., NIAID grant AI142263 to M.C.J., European Research Council (ERC, https://erc.europa.eu/) under the European Union's Horizon 2020 research and innovation programme (ERC-2014-CoG 648432 – MEMBRANEFUSION), Medical Research Council (https://mrc.ukri.org/) MC_UP_1201/16, Deutsche Forschungsgemeinschaft (https://www.dfg.de/) grant BR 3635/2-1 to JAGB, Austrian Science Fund (FWF, https://www.fwf.ac.at/en/) grant P31445 to FKMS. Molecular dynamics simulations were performed on the NCSA Blue Waters supercomputer, supported by the National Science Foundation grant number ACI-1548562. This work used the Extreme Science and Engineering Discovery Environment (XSEDE), which is supported by National Science Foundation grant number ACI-1548562. The funders had no role in study design, data collection and analysis, decision to publish, or preparation of the manuscript.

**Competing interests:** The authors have declared that no competing interests exist.

## Author summary

The structural polyprotein Gag is conserved among all retroviruses and mediates virus assembly via oligomerization into incomplete lattices that are stabilized by dimeric, trimeric and hexameric contacts. Despite a high degree of conservation at the secondary and tertiary structure level, the quaternary interactions between the CA domains of retroviral Gag vary. Recently, the small cellular molecule IP6 was identified as an assembly co-factor of the lentivirus HIV-1. To better understand the structural variability of retroviruses and to determine if IP6 is an assembly cofactor of other lentiviruses we determined the structure of the HIV-1 related retrovirus EIAV. Using cryo-electron tomography and subtomogram averaging, in vitro assembly, mutation analysis, and molecular dynamics simulations, we determined and characterized the structure of the EIAV immature lattice. Furthermore, we found that IP6 is an assembly cofactor of EIAV, and other lentiviruses.

## Introduction

Assembly of the retrovirus particle typically takes place at the inner leaflet of the plasma membrane (PM) and involves the formation of a curved lattice by the structural multidomain protein Gag. This so-called immature lattice, which is attached to the PM via the N-terminal matrix (MA) domain of Gag, can be viewed as a collection of Gag hexamers connected with each other by dimeric and trimeric interfaces. Immediately after, or concomitant with, budding away from the cell, the virus particle goes through a process called maturation in which the viral protease cleaves Gag, releasing the CA domain, which in turn goes on to form the mature lattice. In the course of maturation, the two independently folded halves (or sub-domains) of CA ($CA_{NTD}$ and $CA_{CTD}$), rearrange to create a new lattice of hexamers, but with an entirely new set of interactions [1,2]. Only upon complete maturation is the virus particle infectious.

Equine infectious anemia virus (EIAV) belongs, like HIV-1, to the lentivirus genus of retroviruses [3]. EIAV Gag and HIV-1 Gag share only 30% amino acid sequence identity but have an overall similar domain arrangement (S1A and S1B Fig), with the canonical Gag domains MA, capsid (CA), nucleocapsid (NC), and an unstructured C-terminal domain mediating the late stages of budding. In addition, these two retroviral Gag proteins include a short segment of polypeptide, in HIV-1 termed SP1 ("spacer", here generically called "SP"), that is critical for formation of the immature lattice [4,5]. SP is similar to domains in the Gag proteins of the alpha-retrovirus Rous sarcoma virus (RSV) [6–10] and of the gamma retrovirus murine leukemia virus (MLV) [11] in that it folds into a six-helix bundle (6HB) at the base of the CA domain of the Gag hexamer. The 6HB stabilizes, and indeed may nucleate, assembly of the hexamer. Mutations in SP result in assembly defects and loss of infectious viral particle formation [5,12,13]. SP acts like a switch; its unfolding or proteolytic removal leads to mature CA assembly [14]. Maturation inhibitors target the $CA_{CTD}$-SP junction in HIV-1, preventing unfolding of SP, effectively stopping the immature virus from transitioning into an infectious particle [15,16]. Acting in conjunction with SP is the small cellular molecule inositol hexakisphosphate (IP6), which was shown recently to be an HIV-1 assembly cofactor [17–19]. The binding of IP6 to two rings of six lysine residues in the Gag hexamer, one created at the $CA_{CTD}$-SP interface and the other created by the major homology region (MHR), promotes immature Gag hexamerization and thus virus particle assembly. After the viral protease ablates this immature binding site, IP6 is inferred to be released and then to interact with a ring of six arginine residues in the $CA_{NTD}$, thereby enhancing the formation of the mature hexameric HIV-1 CA lattice and promoting infectivity [2,17,18]. In cells IP6 is synthesized by conversion

of IP5 to IP6 by the enzyme inositol-pentakisphosphate 2-kinase (IPPK). Depletion of IP6 from mammalian cells via CRISPR/Cas9 knock-out of the IPPK gene results in a greater than 10-fold reduction in the production of infectious HIV-1 particles [17].

For several retroviruses both immature and mature lattices can be assembled *in vitro* from purified proteins in the absence of membranes [20]. The protein structures of the resulting virus-like particles (VLPs) accurately mimic the structures of authentic particles formed in cells [15,21]. *In vitro* assembly is carried out by mixing purified, E. coli-expressed, full-length or truncated Gag or CA proteins with buffers and small molecules. Immature assembly typically requires the presence of nucleic acid, a requirement often fulfilled by DNA oligonucleotides. That IP6 promotes HIV-1 assembly was first demonstrated in such an *in vitro* system [22]. *In vitro* assembly systems also have been used extensively to study retroviral structure [15,23–25].

Cryo-electron tomography (cryo-ET) and subtomogram averaging can provide high-resolution structural insights into irregular pleomorphic assemblies [26, 27]. Recently, these techniques have been used to define the structure of the immature HIV-1 lattice to high resolution, both within authentic immature virus particles and with VLPs assembled *in vitro* [14,15]. These studies visualized structural features that regulate HIV-1 assembly and maturation, and increased the understanding of the mode of action of maturation inhibitors [15]. Cryo-ET and subtomogram averaging comparisons of immature Gag lattices of retroviruses from three different genera—alpha-retroviruses, beta-retroviruses and gamma-retroviruses—revealed that despite their high degree of tertiary CA structure conservation, viruses from these three genera adopt different quaternary $CA_{NTD}$ arrangements [1,6,11,21]. These findings thus raise questions about the structural conservation of assembly and maturation mechanisms among the different retroviruses.

Here we report the establishment of an *in vitro* assembly system for EIAV. As found for HIV-1, IP6 promotes assembly of EIAV *in vitro*. Using cryo-ET and subtomogram averaging we have generated structures to below 4Å resolution, revealing conserved and unique structural elements in comparisons between these two lentiviruses. The EIAV CA-SP junction forms an IP6 binding site that interacts with IP6 in a fashion similar to that for HIV-1. This conclusion is supported by *in vitro* assembly, mutational analysis, and all atom molecular dynamics (MD) simulations. Furthermore, our results agree with previous inferences that the retroviral $CA_{NTD}$ and $CA_{CTD}$ are independent structural entities that act autonomously in regulating immature virus particle diameter and curvature. Nevertheless, immature assembly is defined by a set of conserved structural interactions leading to dimeric, trimeric, and hexameric interfaces that are largely determined by the $CA_{CTD}$-SP region, independent of overall particle morphology. In contrast, the immature $CA_{NTD}$ is predominantly stabilized by trimeric inter-hexamer interactions, which remain stable even upon strong distortion of intra-hexameric interactions.

## Results

### *In vitro* assembly of EIAV VLPs

To generate VLPs for cryo-ET and biochemical and mutational analysis, we purified EIAV Gag protein lacking the C-terminal p9 domain, here referred to as Gag for simplicity. The equivalent p6 domain of HIV-1 Gag is not necessary for *in vitro* assembly of VLPs [20]. We also purified a Gag protein lacking both the C-terminal p9 domain and the N-terminal MA domain (GagΔMA) (S1A Fig). These two proteins were screened for their ability to form immature VLPs by *in vitro* assembly. The full length EIAV Gag protein, i.e. with the p9 and MA domains, was insoluble and therefore not studied further. Assembly conditions were assayed by subjecting EIAV Gag and EIAV GagΔMA to different salt concentrations, DNA oligonucleotide

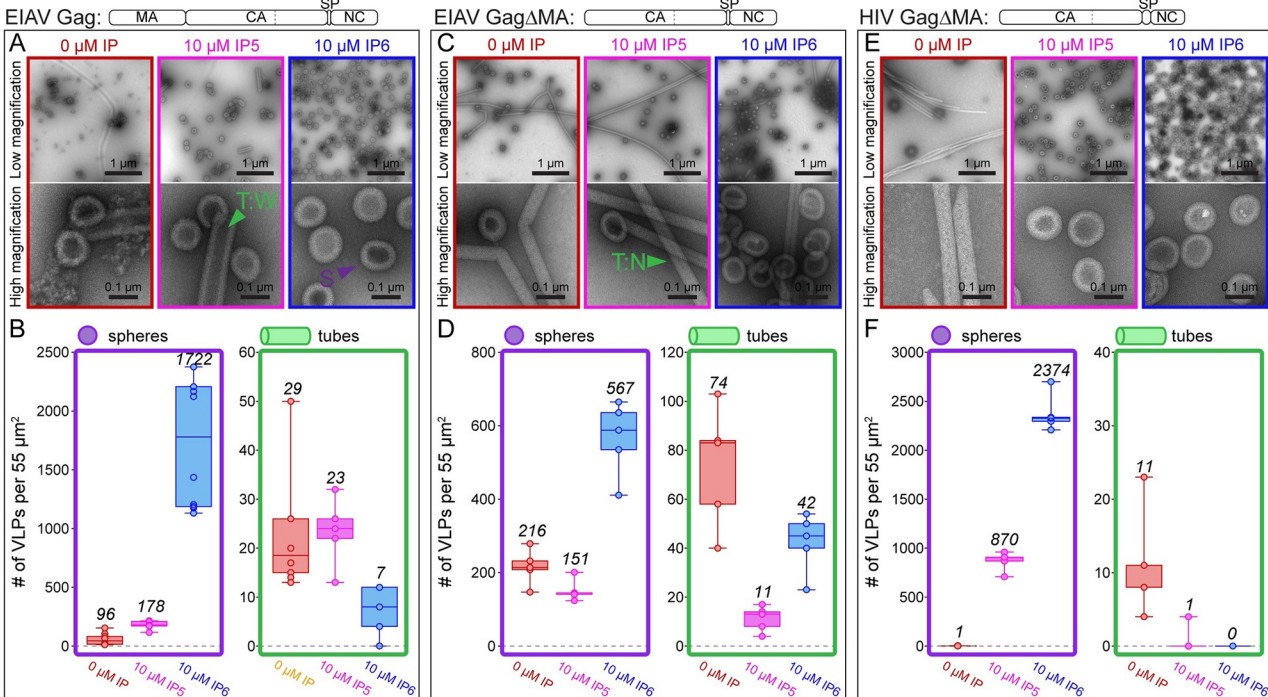

**Fig 1. Effect of IP5 and IP6 on in vitro assembly of EIAV Gag, EIAV GagΔMA, and HIV GagΔMA.** (A,C,E) Representative low and high magnification images of respective proteins assembled in the absence (red) or presence of 10 μM IP5 (pink) or 10 μM IP6 (blue) at pH 6. Examples of wide tubes (T:W), narrow tubes (T:N), and spheres (S) are indicated by green and purple triangles. (B,D,F). The number of VLPs (spheres-purple, tubes-green) per 55μm$^2$ for no fewer than five representative images for each condition. Center lines show the medians; box limits indicate the 25th and 75th percentiles as determined by R software; whiskers extend to minimum and maximum values; data points are plotted as circles. Please note the different Y-axis ranges for the bar chart plots in B, D and F. The low magnification images are representative of the distribution of spheres and tubes assembled under different conditions, while the high magnification images were selected to illustrate their morphology. The mean value of counted particles is given in italics in the bar charts.

templates, pH values, temperatures, and inositol phosphates. EIAV GagΔMA assembly was most efficient at 100mM NaCl, but surprisingly EIAV Gag assembly occurred efficiently at high ionic strengths, e.g. 450mM NaCl at pH 6 (Fig 1A and 1B) and pH 8 (S2A and S2B Fig).

For both of these proteins the N-terminus has an ectopic (non-viral) serine from the purification procedure. GagΔMA beginning with the natural proline, which is found at the N-terminus of all retroviral CA proteins, assembled into tubes with the same diameter as those formed for GagΔMA with the ectopic serine in the absence of IP6 (Fig 1 and S2C Fig). In the presence of IP6, GagΔMA with the natural proline formed multilayered spheres while GagΔMA with the ectopic serine formed single layered spheres. The presence of NC was required for assembly except at high protein and IP6 concentrations, as had been demonstrated previously for HIV-1 [17] (S2D Fig). For all NC containing proteins screened, assembly was dependent on the presence of a fifty-nucleotide single stranded DNA oligonucleotide composed of twenty-five repeats of GT (GT$_{25}$). Interestingly, Gag formed primarily spherical particles while GagΔMA formed both tubes and spheres. To our knowledge, until now, no methods for *in vitro* assembly of EIAV Gag proteins into VLPs had been reported.

### Effect of IP5 and IP6 on EIAV assembly

In the absence of IPs, at pH 6 Gag formed predominantly spheres (~100 spheres and ~20 tubes per defined area), and GagΔMA protein formed long narrow (~35nm) tubes and regular

spheres (~100nm) (Fig 1A–1D). At pH 8 Gag formed predominantly spheres, and GagΔMA yielded both wide (~70nm) tubes and regular spheres (S2A and S2B Fig). Based on these observations we predicted that the narrow GagΔMA tubes at pH 6 represent a mature lattice, consistent with assembly for the corresponding HIV-1 proteins [17], and that the wide tubes and spheres represent immature lattices, consistent with what is known about immature assembly of MPMV, RSV, and HIV-1 [6,15,24]. However, as described below, this prediction proved to be partially incorrect.

In vitro assembly of HIV-1 Gag and Gag truncations is stimulated both by IP5 and IP6 [17]. The myo- inositol form of IP6, with 5 equatorial phosphates and 1 axial phosphate, is the most prominent form of inositol [28]. IP5 lacks the 2' axial phosphate (S1C Fig). For HIV-1 we previously showed that while IP5 is able to promote GagΔMAΔp1Δp6 (referred to here as GagΔMA, see S1A Fig for the construct) assembly into immature VLPs, it does so less well than IP6. The EIAV Gag major homology region (MHR) and CA-SP junction region contain lysine residues at the same positions as those employed by HIV-1 for IP6 interaction (S1B Fig). Direct comparison of VLP formation by EIAV Gag, EIAV GagΔMA, and HIV-1 GagΔMA demonstrated that the ability of IP5 and IP6 to promote immature assembly is less pronounced for EIAV than for HIV-1 (Fig 1). For example, IP6 increased the number of round VLPs by 100-fold for EIAV Gag and 3-fold for GagΔMA, but by as much as a 2000-fold for HIV-1 GagΔMA. By contrast, IP5 increased round VLP formation by only 10-fold for EIAV Gag, had almost no effect for EIAV GagΔMA, but resulted in a 1000-fold increase for HIV-1 GagΔMA. Notably, the overall particle number resulting from IP6-enhanced assembly did not differ significantly for EIAV Gag and the equivalent HIV-1 protein GagΔMAΔp1Δp6. The large difference in the fold increase is due to the number of spherical particles that were already observed in the absence of IP6, ~1 and ~100 for HIV-1 and EIAV, respectively. The effect of IP6 on EIAV assembly did not qualitatively differ between pH 6 and 8 (Fig 1A–1D, S2A and S2B Fig). In summary, these results demonstrate that as for HIV-1, EIAV immature assembly is enhanced by IP6, although it appears less dependent on IP6.

To determine if EIAV is sensitive to IP6 depletion *in vivo*, we used the previously described IPPK knock out cell line coupled with relative infectivity assays [17]. Ablation of this gene reduced infectious particle production by 2-fold, much less than the 100-fold observed for HIV-1 (S3A Fig). Western blots demonstrate that the release of EIAV from cells was not significantly different between control and IPPK KO cells (S3B Fig). These results are consistent with the *in vitro* assembly data showing that immature EIAV assembly is enhanced by, but not dependent on, IP6. For HIV-1 the western blots show a significant decrease in virus release from IPPK KO cells compared to control cells. We interpret this to mean that the large reduction in infectious virus particle production for HIV-1 is due to a reduction in immature virus particle release.

## Cryo-ET analysis of EIAV Gag assembly products

To determine the structure of the *in vitro* assembled EIAV VLPs formed by GagΔMA, and to further elucidate the role of IP6 in regulating assembly, we acquired and processed cryo-ET data of spherical and tubular particles assembled in the presence of IP6 at both pH 6 and pH 8, using our previously published approach [15] (S1 Table). Particles formed by Gag (i.e. containing the MA domain) showed a significant degree of clustering upon vitrification, and attempts to perform cryo-ET of these particles therefore remained unsuccessful. In all cases, the EIAV GagΔMA particles displayed an arrangement of their Gag layer (Fig 2A, S4A Fig), that is reminiscent of what was observed in *in vitro* assembled immature HIV-1 or M-PMV

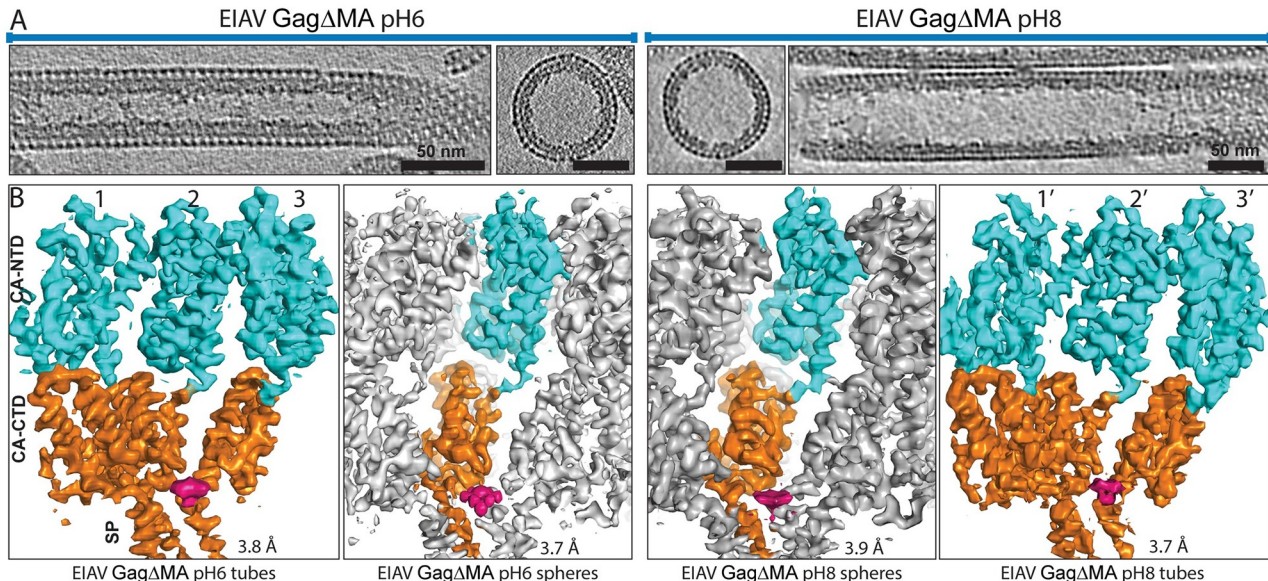

**Fig 2. Cryo-electron microscopy of EIAV GagΔMA assemblies. (A)** Sum of 10 computational slices through gaussian-filtered tomograms containing EIAV GagΔMA tubes and spheres assembled at pH6 and pH8. Protein density is black. Scale bar is 50nm; note the significantly smaller diameter of tubes assembled at pH6. **(B)** Isosurface representations of the subtomogram averages derived from the different EIAV GagΔMA assemblies at pH6 and pH8. In all cases, the CA$_{NTD}$ and CA$_{CTD}$-SP are colored cyan and orange, respectively. The IP6 density is colored in pink. The symmetry-independent copies of CA-SP are denoted with number 1,2,3 and 1',2',3' for the tubes at pH6 and pH8, respectively. For the spheres, only one monomer is colored as all monomers in the hexamer are symmetry-related. In all structures, the helical pitch and densities for larger and several smaller side chains are visible in the EM-density, in good correspondence with the observed resolutions.

tubes and spheres [21,24]. This result was surprising, since HIV-1 and RSV narrow tubes are exclusively in a mature arrangement [25,29,30] and we had therefore predicted that the narrow EIAV tubes would form a mature lattice. In order to obtain more detailed insights into the molecular interactions of GagΔMA in the assembled particles, we performed subtomogram averaging.

The density maps obtained for spheres, and wide and narrow tubes assembled at pH 8 and pH 6, respectively, revealed a conserved order of densities corresponding to the CA$_{NTD}$ at the particle surface, followed by the CA$_{CTD}$, and then the NC-nucleic acid complex (Fig 2, S4A Fig). At the base of CA, just distal to the NC-nucleic acid density, a density consistent with a 6-helix bundle (6HB) was resolved, likely corresponding to the last residues of CA$_{CTD}$ and the first residues of SP, similar to previous maps determined for immature HIV-1 VLPs [15]. In the center of the hexamer, above the 6HB region, an additional density was present, which we interpret to be a bound IP6 molecule. The density is similar in strength to that of the surrounding protein, suggesting that the majority of hexamers contain a bound molecule. No ordered density was observed for NC in any of the structures. In the tubes and spheres assembled at pH6, disordered densities were also observed at the base of the CTD dimer interface, i.e. at the contact points between hexamers (S5 Fig).

The resolution of the different EIAV GagΔMA structures was determined by Fourier-shell correlation (FSC) and in all cases was below 4 Å (S4B Fig and S1 Table). This allowed us to build and refine a model of the EIAV CA and SP domains placed into the experimental electron microscopy densities in the spheres, narrow tubes, and wide tubes. (Fig 3, S1 Movie). We then used the model for further analysis and validation by Molecular Dynamics simulations (S6A and S6B Fig).

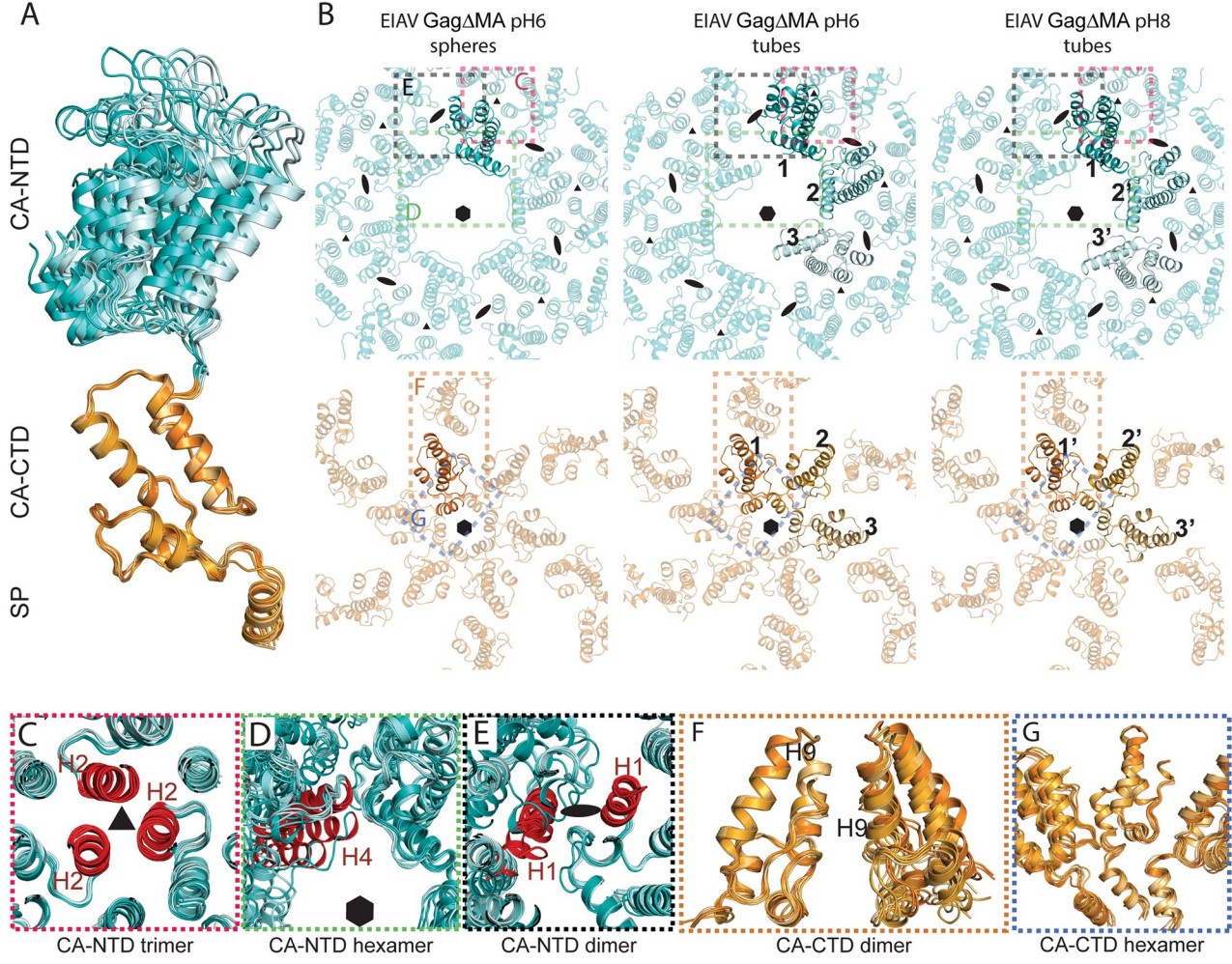

**Fig 3. Structural variability of immature EIAV GagΔMA assemblies. (A)** Superimposition of the seven conformations adopted by CA in spheres and tubes. All monomers are aligned on the $CA_{CTD}$. For the monomer in the spheres the $CA_{NTD}$ and $CA_{CTD}$-SP are colored in cyan and orange, respectively. For the tubular assemblies the symmetry-independent copies (1,2,3 and 1',2',3' for tubes assembled at pH6 and pH8, respectively) are colored pale cyan, aquamarine and teal for the $CA_{NTD}$ and light orange, bright orange and orange for the $CA_{CTD}$. The color coding is identical for all panels. **(B)** The refined models of the different EIAV GagΔMA assemblies are shown as seen from the outside of the virus particle, centered above the CA-SP hexamer. The top panel shows only the $CA_{NTD}$ assembly, the bottom panel only the $CA_{CTD}$-SP assembly. The numbering for the symmetry independent copies is annotated (as described in Fig 2B). The 6-fold, 3-fold and 2-fold symmetry axes are annotated by a hexamer, triangle, and oval respectively. The distortion of the CA hexamer in the tubes at the $CA_{NTD}$ is clearly visible and leads to a separation of the two halves of the immature hexamer. There is significantly less distortion at the $CA_{CTD}$. Colored rectangles indicate regions enlarged in Panels C-G. **(C-G)** In order to show structural similarities and variations within the immature EIAV assembly in tubes and spheres the models from the tubes were triplicated. The symmetry independent monomers in each of the triplicated models were then aligned against the CA monomer determined in the spheres. This allows visualization of the differences that symmetry independent CA monomers can adopt in relation to their neighbors. **(C-E)** All symmetry independent copies have been aligned on the $CA_{NTD}$. Models are shown as seen from outside the virus. **(F-G)** All symmetry independent copies have been aligned on the $CA_{CTD}$. Models are shown in a 90-degree rotation compared to (C-E). **(C)** the trimeric interface stabilizing the interhexameric interactions. This trimeric interface involving helices 2 (colored in red) is rigid and almost no structural changes can be seen. **(D)** Interactions around the hexameric ring are shown. Helices 4 of one monomer of the symmetry independent copies are colored in red to show the large structural variation of the intra-hexameric interactions at the $CA_{NTD}$. **(E)** The dimeric interface between helices 1 (shown in red) is variable as seen by the increasing separation of the helices and the lack of alignment of adjacent monomers. **(F)** The inter-hexameric interactions at the $CA_{CTD}$ are maintained via a dimeric interface involving helices 9 in two adjacent monomers. The distance between the dimeric interface stays fixed, despite small changes in angular relation between the monomers forming the dimeric interface. **(G)** The interactions around the 6HB and the hexameric ring involving residues in the MHR are maintained, but show a variable degree of flexibility to adapt to the varying diameter in the tubes and spheres.

## Structural versatility of EIAV CA domains in immature assembly

Tube morphologies have been observed previously for *in vitro* assembled Gag proteins, where wide tubes were determined exclusively to have an immature lattice [24], and narrow tubes were inferred to have a mature lattice [25,30]. As described above, both wide and narrow EIAV GagΔMA tubes adopted an immature CA arrangement. We therefore aimed to determine the key interactions that form the immature hexameric assembly in EIAV, but still allow the CA domains to accommodate the significant morphological differences between spheres (~100 nm) and tubes of different diameters (~35 nm and ~70 nm).

CA domain monomers within tubes are present in three symmetry-independent copies, due to the inherent two-fold symmetry within the tubular assemblies (Figs 2B, 3A and 3B). Refinement of the model into the independent CA copies resulted in seven structural states of immature EIAV CA-SP (three symmetry-independent CA-SP copies for each of the narrow and wide tubes, respectively, and one CA-SP monomer for the spherical assembly). The root-mean-squared deviation (RMSD) variation between the individual NTD and CTD subdomains of CA was low (0.65±0.16 and 0.55±0.1 Å, respectively), but when aligning the CA domains on their C-termini (Fig 3A), a significant variation of the NTD orientation among these models was observed (RMSD of 9.5 Å, ranging from monomer 1' to 3, as annotated in Figs 2 and 3B). No interactions between the $CA_{NTD}$ and the $CA_{CTD}$ were observed (Fig 3A), indicating that the linker (residues 270–274) connecting the two halves of CA allows them to act as two independent structural entities.

Close comparison of all inter- and intra-hexameric interactions across the three structures showed that three major interaction interfaces (one at the $CA_{NTD}$ and two at the $CA_{CTD}$) need to be maintained in the immature assembly, even with changing particle diameter or assembly morphology (Fig 3B, S2 Movie). A trimeric inter-hexameric interface at the $CA_{NTD}$, involving helix 2 of CA monomers of adjacent hexamers (Fig 3C), is entirely unaffected by the change of the assembly phenotype. In contrast, the different assemblies display highly variable $CA_{NTD}$ interfaces around the hexameric ring (Fig 3D). This also leads to a large structural variability of a potential dimeric $CA_{NTD}$ inter-hexamer interface involving helix 1 (with a maximal RMSD of 11.3 Å between the different helix 1 conformations, Fig 3E). Specific, defined interfaces therefore do not seem to be required at these positions for the stabilization of the immature lattice. At the CTD, a hydrophobic dimer interface involving helices 9 establishes inter-hexameric interactions and remains intact irrespective of the assembly phenotype (Fig 3F). The $CA_{CTD}$ dimer interface appears to accommodate curvature changes between the different hexamers, as no increase in distance of the interface is observed, but rather a slight rotation of the two helices 9 with respect to each other (Fig 3F). Such a hydrophobic interface has already been described for other immature retroviral assemblies. Intra-hexameric $CA_{CTD}$ stabilization is maintained by interactions around the hexameric ring involving residues in the MHR and the 6HB (Fig 3G), which remain intact in all assembly morphologies. Taken together these results identify key interaction domains in the EIAV immature lattice. Although intra-hexameric $CA_{NTD}$ interactions can be formed in EIAV, their high structural variability suggests that the $CA_{NTD}$ is not required to establish intra-hexameric interactions, but rather forms a trimer to promote interactions with neighboring hexamers. Intra-hexameric interactions are maintained by the $CA_{CTD}$ and the CA-SP 6HB.

## EIAV and HIV-1 use conserved interactions to form the immature CA-SP lattice

Retroviruses have low CA amino acid sequence similarity, but have highly conserved tertiary folds of their CA domains. Previous reports have compared the quaternary CA structures of

retroviruses from different genera and found them to be different at the $CA_{NTD}$ [6,11,21]. Until now, it was not known if immature CA assembly is conserved within the same retroviral genus, lentivirus in this case, nor what role highly conserved residues among lentiviral CA domains might play in establishing important interfaces in the immature assembly. Here we compare the structures of EIAV and HIV-1, two lentiviruses that share 29% and 42% identity between their $CA_{NTD}$ and $CA_{CTD}$, respectively (S1B Fig).

As described above, both tubular and spherical EIAV GagΔMA particles maintain identical key interactions within the immature lattice. We used the model derived from spheres assembled at pH6 for further analysis, in order to compare the interfaces present in immature EIAV with those previously described for HIV-1 [15,31]. Overall, immature assembly in EIAV and HIV-1 is very similar. Mapping the conserved residues onto our structures reveals similarities and subtle differences. EIAV and HIV-1 maintain similar immature assemblies by using residues with equivalent biochemical properties. Of the conserved CA amino acids, the majority are positioned to maintain the tertiary fold of the CA domain, while a minority (~10%) is involved in contacts between CA domain monomers in the immature lattice (Fig 4A, S3 Movie). We predict that the rest are important for maintaining CA contacts in the mature lattice.

In both lentiviruses, the $CA_{NTD}$ is positioned to form extensive inter-hexamer contacts. Of note, both retroviruses have a trimeric $CA_{NTD}$ interface around helices 2 that is structurally similar despite weak sequence conservation. The structural similarity nevertheless suggests that this might be an important interface for maintaining the $CA_{NTD}$ quaternary structure (Fig 4B left). The distance across the helix 1 $CA_{NTD}$ dimer interface (as also described in HIV-1 previously [21]) is variable between EIAV and HIV-1, suggesting that this interface is not conserved for immature lentivirus CA assembly. In the EIAV lattice, helix 1 begins at residue G143, and the upstream residues 140TPR142 appear less ordered and cannot be modeled into an alpha-helix. This differs in HIV-1, where the corresponding residues 148SPR150 are part of helix 1 (Fig 4B right). The shorter helix 1 in immature EIAV agrees with a previous NMR study that reported the first residues of helix 1 to become ordered only upon maturation and beta-hairpin formation [32].

EIAV hexamers are linked at the $CA_{CTD}$ via a dimeric interface involving hydrophobic residues F308 and L309 in helix 9 (Fig 4C). This interface corresponds to the dimer interface in HIV across helix 9 maintained by residues W316 and M317. Conserved residues in the MHR and helix 11 in both EIAV (R278, E336) and HIV-1 (R286, E344) are positioned to establish an interaction that stabilizes the immature hexameric assembly at the $CA_{CTD}$ (Fig 4D, left).

The structural arrangement formed by the 6HB and the base of $CA_{CTD}$ is similar to the hexameric assembly unit described in HIV-1, where the MHR, the $CA_{CTD}$ base, and the residues in the hinge connecting helix 11 and the CA-SP1 helix are packed together. In HIV-1 this assembly unit is stabilized by a three-way interaction involving D329, P356 and H358 (Fig 4D, right). Mutation of any one of these three amino acids results in a complete switch from an immature to mature assembly phenotype [31]. In vivo, mutating these residues results in a severe loss of particle production [33]. In EIAV this interface is less tightly packed, consisting of residues E321, T348 and Q350 (Fig 4D, right). In order to determine whether this three-way interaction is critical for the immature EIAV lattice, we created Q350 mutants and analyzed *in vitro* assembly of GagΔMA and *in vivo* infectious particle production. The Q350A mutation resulted in a ~20-fold decrease in the number of VLPs in the absence of IP6 compared to WT assembly. IP6 did stimulate assembly of Q350A suggesting that the mutation does not alter the IP6 binding site. *In vivo*, the mutation reduced the production of infectious virus particles by less than two-fold compared with wild type (S7 Fig). MD simulations revealed that for HIV-1 the contact occupancy between H358 and D329 or P356 is high (95% and 94%, respectively).

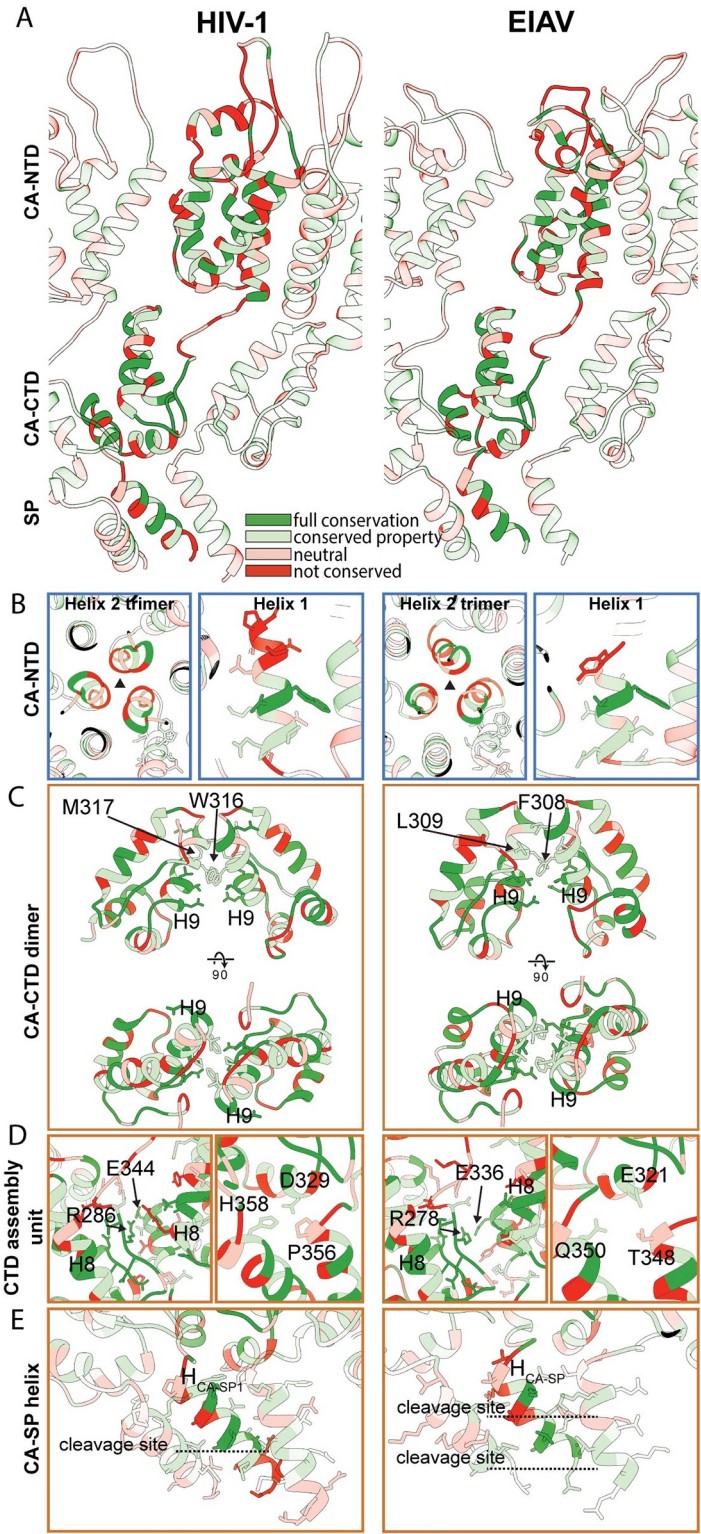

**Fig 4. Conserved structural interactions in EIAV and HIV-1.** Comparison of structural features in EIAV and HIV-1. The HIV-1 CA-SP1 model derived from HIV-1 GagΔ16–99Δp6 (pdb 5l93, referred to as ΔMACANCΔp6 in [15]) is shown on the left. The EIAV CA-SP model derived from EIAV GagΔMA is shown on the right. S3 Movie shows a guided tour of this comparison. **(A)** Side view on the CASP lattice of EIAV and HIV-1 CASP. One monomer is highlighted, surrounding monomers of the lattice are shown with reduced opacity. The residues are colored according to the conservation between the two viruses. The color legend is indicated in panel (A). **(B)** Interactions and structural features in the CA_NTD. The trimeric interface in EIAV and HIV-1 is similar (Left). The Helix 1 in EIAV is shorter than

in HIV-1 (Right). The extent of helix 1 in EIAV approximately corresponds to the conserved residues in HIV-1. **(C)** Comparison of the dimeric $CA_{CTD}$ interface. Both lentiviruses use hydrophobic residues in helix 9 to stabilize the dimeric interface. F308/L309 and W316/M317 are annotated in EIAV and HIV-1, respectively. **(D)** Conserved residues in the MHR and helix 11 contribute to interactions around the hexameric ring to stabilize the immature CA assembly (Left). In HIV-1 residues D329, P356 and H358 form an important three-way interaction linking the $CA_{CTD}$ base and the CA-SP1 helix of two adjacent CA monomers to each other. The equivalent residues in EIAV are E321, T348 and Q350 (Right). **(E)** The EIAV CASP 6HB is shorter than its counterpart in HIV-1. In EIAV and HIV-1 the CA-SP cleavage site is located within the helix, while the SP-NC cleavage site is located below the helix. Proteolytic cleavage sites are annotated by dashed lines.

These values are significantly greater than the contact occupancy for EIAV at the same interface, Q350 and E321 or T348 (61% and 23%, respectively). Taken together, these results indicate that this junction in EIAV might be a less critical regulator of immature assembly than has been described for HIV-1.

In EIAV the last eight residues of the $CA_{CTD}$ (TTKQKMML) and all five residues of SP (LAKAL) (S1B Fig) form a helix that arranges into a 6-helix bundle in the immature lattice (Figs 2B, 3B and 4E). Identical to HIV-1, in EIAV 12 Lysine residues in the MHR (K282) and in the CA-SP (K351) project towards the center of the hexamer. The density for the CA-SP helix stops at residue L359, which is the cleavage site between SP and NC, making the 6HB approximately one turn shorter than its HIV-1 counterpart (Fig 4E). In HIV-1, residues in the 6HB that are critical for assembly are A360 and L363 [15,31,33]. In EIAV, interactions to stabilize the 6HB seem to be similarly established by hydrophobic residues (M352, L355 and A356) (Fig 4E, S1B Fig).

## IP6 has a conserved structural role in immature lentivirus assembly

The density we observed in the center of the EIAV hexamer at the level of the $CA_{CTD}$-SP junction is in an identical position to that previously reported for HIV-1 VLPs and immature viruses [15,21]. The shape and size of the IP6 density is similar in tubular and spherical particles. The better-resolved density in the map from GagΔMA spheres assembled at pH6 has a strikingly similar shape and size to the density observed for IP6 in the recently obtained co-crystal structure of HIV-1 CACTD-SP with IP6 [17]. In our electron microscopy map, densities for the individual phosphate groups were visible and hence allowed fitting of IP6 in its myo-conformation, with one axial and five equatorial phosphate groups (Figs 2B and 5A). Identical to HIV-1, in EIAV IP6 is coordinated by the above described 12 lysines in the MHR and the CA-SP bundle, indicating that IP6 is bound in immature lentivirus CA assemblies in a conserved manner.

## Mutation of IP6-interacting amino acids alters assembly and infectious virus particle production

For HIV-1, mutation of the IP6-interacting amino acids results in both a loss of IP6 enhanced assembly *in vitro*, and a loss of infectivity *in vivo* [17]. For EIAV, mutation of either K282A or K351A resulted in a near complete loss of infectivity (Fig 5B). To further confirm the role of the lysine residues in IP6 binding we compared the *in vitro* assembly properties of GagΔMA with two mutant forms of the same protein, K282A and K351A (Fig 5C–5H). The mutations resulted in a loss of IP6-enhanced assembly. Thus, IP6 increased the number of spherical VLPs counted for wild type GagΔMA protein by ~10-fold, but this small molecule gave no enhancement for the K282A and K351A proteins. Interestingly, both of these mutant proteins assembled better than wild type GagΔMA in the absence of IP6. In summary, these results imply that the amino acids K282 and K351 are critical for IP6 interaction *in vitro*, and for production of

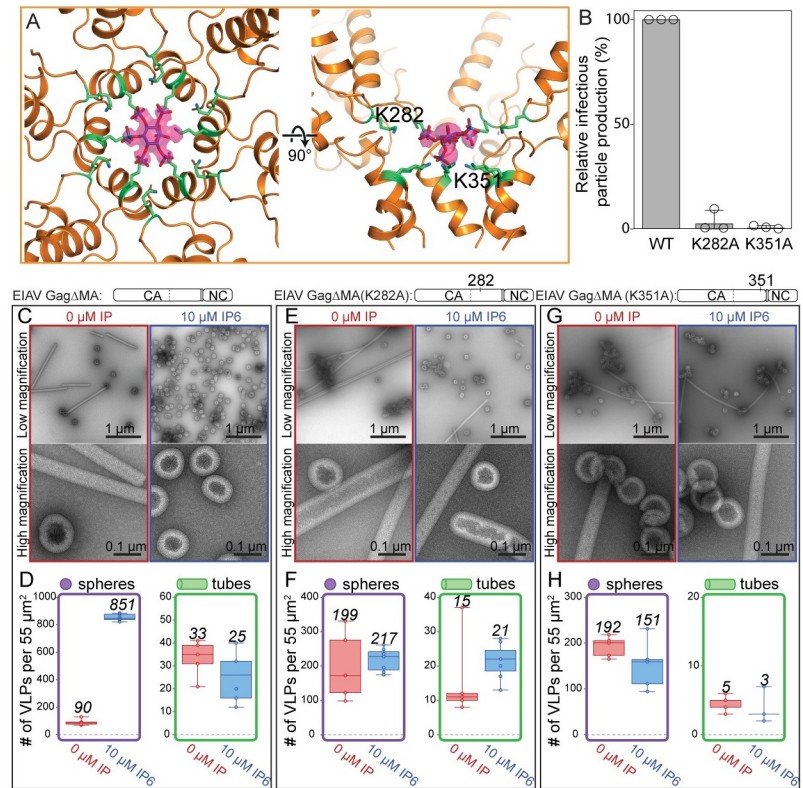

**Fig 5. IP6 stabilizes the immature EIAV CASP lattice. (A)** EIAV CASP and the IP6 molecule are shown as seen from the outside of the VLP and additionally rotated by 90˚. IP6 sits in the center of the hexamer and is coordinated by a ring of six lysines in the MHR (K282) and six lysines in the CASP 6HB (K351). An isosurface representation of the IP6 density is shown in pink. The densities for the individual equatorial and the axial phosphate groups are clearly visualized. The non-occupied phosphate group is caused by the 6-fold symmetry applied during processing and the fact that IP6 can sit in the binding site in 6 rotationally equivalent positions. **(B)** Relative infectious particle production in 293FT cells of VSV-G-pseudotyped provirus of wild type EIAV Gag (WT) and Gag with point mutations. Graphs show the average and standard deviation of three independent experiments; dots show individual data points. **(C,E,G)** Representative low and high magnification images of GagΔMA WT, K282A, and K351A proteins assembled in the absence (red) or presence (blue) of 10 µM IP6 at pH 6. **(D,F,H)** The number of VLPs (spheres-purple, tubes-green) per 55µm² for no fewer than five representative images for each condition. Center lines show the medians; box limits indicate the 25th and 75th percentiles as determined by R software; whiskers extend to minimum and maximum values; data points are plotted as circles. The mean value of counted particles is given in italics in the bar charts.

infectious virions *in vivo*, and that the binding site for IP6 is highly conserved between HIV-1 and EIAV.

## IP6 does not alter 6-helix bundle stability

Previously, MD simulations of HIV $CA_{CTD}$-SP demonstrated that the presence of IP6 provides a dramatic stabilizing effect on the 6HB, with overall C-alpha RMSD values of 2.5 and 4 Å for bound and unbound IP6, respectively [17]. In the absence of IP6, the 6-helix bundle collapsed rapidly during simulation time. In HIV-1, through contacts with K290 and K359, IP6 stabilized this region. EIAV differs from HIV in this respect, showing no dependence on IP6 for six helix bundle stability over simulation time, with C-alpha RMSD values of ~1.5 for both bound and unbound IP6. While IP6 binding to the two rings of six lysine residues (K282 and K351) formed by $CA_{CTD}$-SP hexamerization is stable, it is dispensable for maintaining the stability of

the hexamer structure over simulation time (S6 Fig, S4 Movie). The latter is a result of the greater stability of the side-chain interactions between helices of the six-helix bundle of EIAV compared to HIV-1; remarkably most of the interactions are hydrophobic in nature. These observations are consistent with the *in vitro* assembly data which show that EIAV, but not HIV-1, can form immature VLPs in the absence of IP6, and that immature assembly for both is stimulated by IP6.

### Effect of IP6 on other lentiviruses

Sequence comparisons of the MHR and CA-SP junction show conservation of IP6-interacting lysine residues for HIV-1, EIAV, simian immunodeficiency virus (SIV), HIV-2, feline immunodeficiency virus (FIV), and bovine immunodeficiency virus (BIV) (Fig 6A). Thus, we predict that all lentiviruses form a similar binding pocket, and that IP6-enhanced assembly is conserved among lentiviruses. To test this hypothesis, we purified GagΔMA proteins for HIV-2, SIV, FIV, and BIV, and determined if their assembly properties are altered by IP6 (see construct diagrams in 6B-E). To our knowledge, no *in vitro* assembly conditions had been established previously for any of these viruses, requiring us first to screen for conditions that support assembly. In the absence of IP6, all four retroviral proteins failed to form VLPs, as analyzed by negative stain TEM. These results differ from those for EIAV, which forms some immature VLPs, and for HIV-1, which forms mature tubes in the absence of IP6 (Fig 1). For all four other lentiviruses, in the presence of IP6, spherical VLP assembly consistent with an immature lattice was observed (Fig 6B–6E). VLP diameters fell into two groups. HIV-2 and SIV had average diameters similar to those of HIV-1 and EIAV, greater than 100nm (S8B Fig). FIV and BIV had significantly smaller diameters of ~80nm. FIV and BIV VLPs, while abundant, were also less regular. The observation that assembly of all of these lentivirus Gag proteins is stimulated by IP6 suggests that IP6 plays an evolutionarily conserved role in immature virus particle assembly for this genus of retroviruses.

## Discussion

### Comparison of immature EIAV and HIV-1 assembly

We have identified *in vitro* assembly conditions for EIAV Gag proteins that result in VLPs of sufficient quality and abundance for cryo-ET structure determination. Three VLP morphologies were observed: spheres, narrow tubes, and wide tubes. Subtomogram averaging revealed the lattice of all three structures to sub-4Å resolution; surprisingly, all three were immature.

### The immature Gag lattice can adopt a range of morphologies

The spatial separation of EIAV CA$_{NTD}$ and CA$_{CTD}$, and the lack of strong interactions between them, allow these two halves of CA to behave almost as independent entities, as suggested previously for other retroviruses [6,21,34]. For example, assembly of an HIV-1 CA mutant led to immature-like tubes [35], where the CA$_{NTD}$ adopted an artificial p2 lattice with four two-fold positions, while the CA$_{CTD}$ was still arranged into an immature hexameric lattice. Other studies showed that the CA$_{CTD}$ is sufficient to form the protein-protein interactions needed to assemble the immature lattice [36], while the CA$_{NTD}$ regulates curvature and hence size of the virus particle [1]. For example, IP6-induced assembly of HIV-1 CA-SP protein leads to spherical VLPs, while IP6-induced assembly of CA$_{CTD}$-SP (lacking the CA$_{NTD}$) leads to a flat immature CA$_{CTD}$-SP lattice [17]. Our present results extend these observations by illuminating in detail the different conformations that the CA$_{NTD}$ can adopt in order to form an immature lattice, even at higher curvatures as seen in the narrow tube assemblies at pH6. In immature

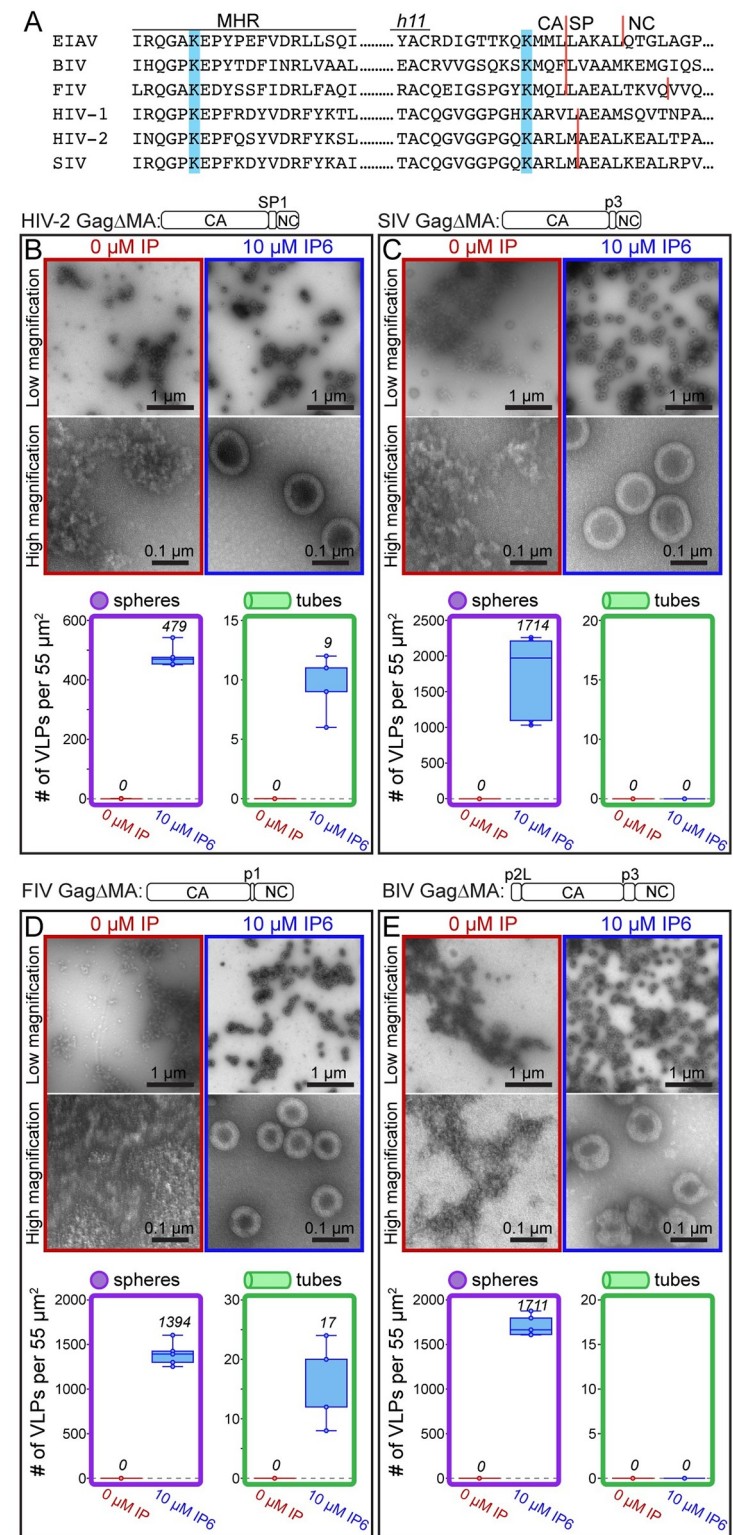

**Fig 6. The effect of IP6 on other lentiviruses. (A)** Comparison of the MHR sequence and the CASP junction of lentiviruses. Blue bars indicate the location of known (EIAV and HIV-1) and predicted IP6 interacting lysine residues. **(B-E)** In vitro assembly results of HIV-2, SIV, FIV, and BIV Gag constructs without (red) and with (blue) IP6 at 22˚C. **(B, C)** HIV-2 and SIV assembly was done in 50 mM Tris pH 8, 100 mM NaCl, with GT25 oligo by dialysis. **(D)** FIV assembly was done in 50 mM Bis-Tris propane, 150 mM NaCl, GT50 oligo by dilution. **(E)** BIV assembly was done in 50 mM MES pH 6.5, 100 mM NaCl, GT50 oligo by dilution. The mean value of counted particles is given in italics in the bar charts.

EIAV the building block of immature assembly at the $CA_{NTD}$ is defined by trimeric inter-hex-amer interactions. A similar trimeric interaction is also observed in the immature HIV-1 lat-tice (Fig 4B), although the residues within helix 2 at this trimer interface are not conserved between EIAV and HIV-1 (S1B Fig). Formation of a trimeric interface is important in HIV-1 assembly: defects in immature HIV-1 Gag assembly are caused by mutation of selected resi-dues in a conserved proline-rich loop that is positioned on the trimeric symmetry axis at the top of the $CA_{NTD}$ [37].

In our previous publications we reported on a potential inter-hexameric $CA_{NTD}$ dimer interface in HIV-1 involving Helices 1, and speculated that this interface might play a role in immature lattice assembly [21]. Here we show that in EIAV such a dimeric interaction is not involved in determining the immature lattice, as the distance between helices 1 is highly vari-able between the different assembly morphologies (Fig 3E). Further experiments are therefore necessary to confirm whether the proposed dimeric interaction in HIV-1 is of relevance in determining the immature lattice.

At the $CA_{CTD}$ the intra- and inter-hexamer interactions in EIAV remain highly similar, despite the variable curvature we have observed in the different assembly morphologies. This further supports previous observations that the $CA_{CTD}$ plays the dominant role in establishing relevant lateral protein-protein interactions in the immature lattice. In immature EIAV the inter-hexamer interactions at the $CA_{CTD}$ are, as reported for other studied retroviruses, estab-lished by a hydrophobic helix 9 dimer interface [6,15,38,39]. It is likely that such a hydropho-bic interface between hexamers at the $CA_{CTD}$ allows for enough flexibility in immature retroviral assembly to accommodate the curvature changes imposed by the varying CA-NTD interactions.

## IP6-enhanced immature lentivirus assembly

Remarkably, in EIAV and HIV-1, IP6 is coordinated by the same two rings of highly conserved lysines in the MHR and the top of the $CA_{CTD}$-SP bundle. This suggests that this defined struc-tural coordination of IP6 by 12 conserved positively charged residues is important for lentivi-rus assembly and stability.

The effects of IP6 on EIAV and HIV-1 differ qualitatively and quantitatively. For HIV-1, in the absence of IP6, Gag protein assembles into mature tubes, while in the presence of IP6 Gag assembles into immature spheres. By contrast, *in vitro* EIAV Gag forms immature virus parti-cles both with and without IP6. We predict that the difference in IP6 dependence of these two lentiviruses is due to the effect of IP6 on 6HB formation. Supporting this hypothesis, in MD simulations the HIV-1 6HB collapses in the absence of IP6 but remains stable in its presence. By contrast, in MD simulations the EIAV 6HB was stable both with and without IP6 likely due to additional side chain contacts between helices in the six-helix bundle.

For HIV-1 *in vivo*, genetic ablation of IPPK, the cellular protein that converts IP5 to IP6, results in dramatic reduction in infectious particle production. In contrast, the effect of the IPPK knockout on EIAV was weak, consistent with the less strict requirement for IP6 in assembly *in vitro*. One possible explanation is that EIAV can use IP5 as an alternative to IP6; however, our observation that IP5 did not stimulate EIAV Gag assembly *in vitro* argues against this model. A second possibility is that another small molecule can promote EIAV assembly. Because of the high positive charge of the IP6 binding site, any hypothetical molecule that mimics IP6 should also have a high negative charge density. The high degree of conservation of the IP6 binding site between EIAV and HIV-1 argues against this model.

A third possible explanation is based on the observation that EIAV infectivity is signifi-cantly lower than that of HIV-1 [40]. We might be observing a basal level of IP6-independent

lentiviral assembly in our cell-line model, which is detected for EIAV relative to the low level of infectivity in control cells, but not for HIV-1 relative to the high level of infectivity in control cells. Conceivably, if infectious particle production for EIAV in control cells were more efficient, the effect of IP6 might be more pronounced. We consider this third explanation to be the most likely.

### IP6 enhanced assembly of other lentiviruses

The role of IP6 as an assembly cofactor apparently is conserved among most lentiviruses, given our findings that *in vitro* assembly of HIV-1, HIV-2, SIV, FIV, and BIV Gag proteins also is dramatically stimulated by IP6. In contrast to EIAV, the other lentiviruses did not assemble in the absence of IP6, suggesting that they have a stricter requirement for the molecule. This observation perhaps is not surprising since EIAV is the most distantly related to HIV-1 of the viruses tested. We are continuing to study these diverse lentiviruses *in vivo* to determine how IP6 depletion affects assembly, budding, and infectivity. It will be critical to determine if IP6 also influences mature assembly of these lentiviruses in the way that it does for HIV-1.

### Conclusion

In summary, EIAV is only the second retrovirus, after HIV-1, to have its immature Gag lattice determined to this level of resolution by cryo-ET and subtomogram averaging. Comparison of these two structures allowed us to identify conserved and variable structural features that determine immature assembly in both viruses. The overall similarity of the interactions that determine immature assembly in EIAV implies that EIAV can be a valuable model for obtaining a deeper understanding of how virus assembly is regulated within the lentivirus genus. Our structures allow for a detailed comparison of the position of conserved and variable residues in interfaces within the EIAV and HIV-1 CA domain, providing a detailed view of the mechanisms these viruses employ to form stable virus particles. Our data clearly show that both EIAV and HIV-1 coordinate IP6 via a pair of lysine rings that are identically positioned within the immature Gag assembly. The identification of the IP6 binding site in EIAV, and the accompanying assembly data for HIV-2, SIV, FIV, and BIV, provide further support to the model that IP6 is a conserved lentiviral assembly cofactor.

## Methods

### *In vitro* assembly

All EIAV Gag constructs were based on a pRE/EIAV Gag expression vector provided by Eric Freed, and cloned into respective vectors using standard molecular cloning methods. Unless otherwise noted, all proteins were purified using the SUMO-tag system [41] as previously described in [17], and stored at -80˚C in storage buffer (500 mM NaCl, 20 mM Tris-HCl pH 8, 2 mM TCEP). Mutations were generated by sub-cloning synthesized DNA harboring the stated point mutation (gBlocks purchased from Integrated DNA Technologies) into the respective vector. Native GagΔMA protein was purified as previously described for RSV GagΔMBD [42].

*In vitro* assembly was performed by dialysis described briefly here. Protein at 50 μM was mixed with 10 μM GT25 oligo. IP5 or IP6 was added to a final concentration of 10 μM to both the reaction chamber and the bulk buffer. 30 μL assembly reactions were dialyzed against 2 mL buffer (50mM Tris-HCl pH 8 or 50 mM MES pH 6, 100 mM NaCl, 2 mM TCEP, with or without IP6). Unless otherwise stated, all assembly reactions were performed at 4˚C for a minimum

of 4 hrs. All assembly reactions were adjusted up to a final volume of 200 μl with dialysis buffer before spotting on EM grids. Samples were spotted onto formvar/carbon grids with a coating for negative charge (Electron Microscopy Sciences; FCF200-CU-SP), stained with 2% uranyl acetate solution, and imaged on an FEI Morgagni transmission electron microscope.

GagΔp9 assembly was performed by mixing 50 μM protein in storage buffer to an assembly buffer to a final concentration of 10 μM GT25; 0 μM IP, 10 μM IP5, or 10 μM IP6; 450mM NaCl; with 50 mM MES at pH 6. Assembly reactions were incubated at 22˚C for 1 hr. Following incubation, reactions were diluted to 200 μL (1:5) with the corresponding assembly buffer before spotting onto EM grids as previously described.

We were unsuccessful in identifying conditions for assembly of the mature EIAV lattice. We screened conditions for assembly of purified EIAV CA, but none of the conditions known to promote robust assembly of HIV-1 or RSV CA resulted in EIAV assembly, e.g. presence of high NaCl (500 mM protein; 1 M or 2 M NaCl; at 4˚C, 22˚C, or 24˚C; in 20 mM MES pH 6 or 20 mM Tris-HCl pH 8), high NaPO4 (500 mM protein; 500 mM NaPO4 pH 7 or 1 M NaPO4 pH 7; at 22˚C) or IP6 (250 mM protein; 2.5 M IP6; at 22˚C; in 20 mM MES pH 6 or 20 mM Tris-HCl pH 8), or crowding reagents such as Ficoll 400 (protein; Ficoll 400; at 4˚C or 37˚C; 20 mM MES pH 6.5 or 20 mM Tris-HCl pH 8).

## Cells and plasmids

The HEK293FT cell line was obtained from Invitrogen and maintained in Dulbecco's modified Eagle's medium (Sigma, Cat#D5796-500ML) supplemented with 10% Serum Plus II (Sigma, Cat#14009C-500ML), 2 mM L-glutamine (VWR, Cat#02-0131-0100), 1 mM sodium pyruvate (Corning, Cat#25-000-CI), 10 mM nonessential amino acids (Lonza, Cat#13-114E), and 1% minimal essential medium vitamins (Sigma, Cat#M6895-100mL). The IPPK KO was derived from the HEK293FT cell line as previously described [17].

The proviral HIV plasmid construct consists of pNL4-3 HIV plus CMV-GFP in place of Nef (kindly obtained from Vineet KewalRamani). This HIV construct has Vif, Vpr and Env deleted and has several restriction sites silently added to the Gag gene for cloning purposes. EIAV proviral constructs consisted of the Gag/Pol expression vector pONY3.1 [40] and the GFP reporter vector pONY8.0G [43]. The K282A, Q350A, and K351A versions of pONY3.1 were made by standard cloning methods. The VSV-G construct consists of the coding sequence for VSV-g preceded by the EFa1 promoter [44] and obtained through the NIH AIDS Research and Reference Reagent Program.

## VSV-G-pseudotyped virus production, infection, and western blots

EIAV and HIV viruses were produced by equal PEI transfection of adhered HEK293FTs and the IPPK-KO in 6-well format at 75–80% confluence with 1000 ng of proviral plasmids and the VSV-g expression construct in a 9:1 ratio (900 ng HIV with 100 ng of VSV-g; 450 ng of pONY3.1 or mutants plus 450 ng of pONY8.0G with 100ng of VSV-g). Media from transfected cells was collected two days after transfection by aspiration. This media was then frozen at -80˚C for a minimum of 2 hrs to lyse cells, thawed in a 37˚C water bath, precleared by centrifugation at 3000 x g for 5 min, and supernatant (viral media) collected by aspiration. Viral media was stored at -80˚C and subsequently used for assays.

To compare EIAV and HIV infectious particle production, HEK293FTs were plated in 12-well format and infected by equal volume addition of viral media from HEK293FTs and from the IPPK-KO at low MOI. WT EIAV and mutant (K282A, Q350A, and K351A) EIAV viral media were similarly infected. Cells were collected 48 hrs post-infection. Number of infections, as measured by green fluorescence, was quantified via flow cytometry (Accuri C6,

BD). The number of infections was expressed as a percentage and normalized to WT percentage of infections. Western blots were performed as previously described [45]. The E. coli-expressed EIAV CANC protein was used in preparation of the rabbit antiserum (Cocalico Biologicals).

## Cryo-electron tomography

Cryo-electron tomography sample preparation and data acquisition were performed as described previously [15]. In brief, 10nm colloidal gold (coated either with BSA or conjugated with Protein A) was added to the VLP solutions and 2.5 μl of this solution was then applied to degassed 2/2-3C C-flat grids, that previously were glow discharged for 30 seconds at 20 mA. The samples were vitrified in liquid ethane using a Vitrobot Mark 2 (blot time of 1–2 seconds, with a blot offset of -3 mm) and stored under liquid nitrogen conditions until imaging.

Tilt series acquisition and tomogram reconstruction were performed in an identical manner for all datasets unless stated otherwise. All imaging was done on an FEI Titan Krios, operated at 300 keV, equipped with a Gatan Quantum 967 LS energy filter and a Gatan K2xp direct electron detector using the SerialEM software package [46]. The slit width of the filter was set to 20 eV. Low magnification montages were acquired for search purposes and for defining areas of interest for subsequent high-resolution tomography data acquisition. Prior to tomogram acquisition, gain references were acquired and the filter was fully tuned. Microscope tuning was performed using the FEI AutoCTF software [47].

The nominal magnification for the dataset containing EIAV GagΔMA assembled at pH8 was 105,000x, resulting in a pixel size of 1.35 Å/pixel. The nominal magnification for the dataset containing EIAV GagΔMA assembled at pH6 was 130,000x, resulting in a pixel size of 1.04 Å/pixel. Tilt series were acquired using a dose-symmetric tilt-scheme [48]. The tilt range was from 0˚ to 60˚ and -60˚ in 3˚ steps. Tilt images were acquired as 8k x 8k super-resolution movies of 20 and 21 frames, for the pH8 and pH6 data sets, respectively. The set dose rate was at ~ 2.5 e$^-$/A/sec. Tilt series were collected at nominal defocus between -1.5 and -5 μm and a target dose of ~140 e$^-$/A$^2$. Data acquisition information is also provided in S1 Fig.

## Image processing

The super-resolution movies were aligned on-the-fly on a GPU during data acquisition using the SerialEMCCD frame alignment plugin and tilt series were automatically saved as 2x binned mrc stacks. CTF-estimation was performed using CTFFIND4 [49] on each tilt individually. Images were low-pass filtered according to their cumulative electron dose using exposure filters that were calculated using an exposure-dependent amplitude attenuation function and critical exposure constants (as determined previously by [50]). Prior to further processing, bad tilts (e.g. images that shifted significantly during acquisition or due to a blocked beam at high tilts) were removed.

Tomogram reconstruction of the exposure-filtered tilt images was performed in the IMOD software package [51]. CTF-correction was performed slightly different for the two datasets. For the data of EIAV GagΔMA assembled at pH8 (spheres, and tubes with wide diameter) 2D CTF-correction was initially performed by the "ctf-phase-flip"-program implemented in IMOD [52]. 3D-CTF correction using NovaCTF [53] was then performed at the end of the subtomogram averaging calculations using Z-height coordinates of the aligned subvolumes to define the center of mass in tomograms to obtain a more precise CTF-correction for final unbinned structure calculation.

The data set containing EIAV GagΔMA assembled at ph6 (spheres and tubes with narrow diameter) was corrected with NovaCTF from the very beginning, using no subtomogram Z coordinates for refining the CTF-correction. Updated Z-positions were then again used at the end of the subtomogram averaging computations to regenerate more accurately CTF-corrected tomograms for final unbinned structure calculation.

CTF-corrected tomograms were reconstructed unbinned, as well as 2x, 4x and 8x binned, using either anti-aliasing (in case for the 2D-CTF corrected tomograms) or Fourier cropping (for 3D-CTF corrected tomograms using NovaCTF).

VLPs were identified within the tomograms using the Amira visualization software (FEI Visualization Sciences group). The radii, and center or spline of the VLPs (for the spherical or tubular VLP assemblies, respectively) were determined in Amira using the electron microscopy toolbox [54]. The pH8 and pH6 datasets contained 158 spheres/106 tubes and 175 spheres/152 tubes, respectively. All four datasets were processed independently.

Subtomogram averaging was performed identically for all datasets using scripts derived from the AV3 [55], TOM [56] and Dynamo [57] packages. The only exception was the use of different subvolume dimensions due to the varying pixel size.

Initially, one tomogram (binned 8x) containing both spheres and tubes assembled at pH8 was chosen to generate ab initio starting references. To generate a starting reference for the narrow tubular EIAV GagΔMA assembly (pH6), again one tomogram was chosen. The starting reference from pH8 was used to also align EIAV GagΔMA in spheres assembled at pH6. Subvolumes with a size of approximately $(380)^3$ Å were extracted from the surface of spheres or tubes, respectively, and averaged. Initial angles were assigned according to the geometry of the spheres or tubes. First iterations were performed without applying any symmetry, and only once the structures converged, revealing the inherent 6-fold and 2-fold symmetry for the structures in spheres and tubes, respectively, the symmetry was enforced for two more iterations. In all cases the starting references showed clear densities for EIAV GagΔMA. From this point onwards, symmetry was applied throughout all subtomogram averaging steps.

Two rounds of alignments were performed on 8x binned data. Subsequently, all subvolumes that had converged onto the same position of the lattice were removed using a subvolume-to-subvolume distance cut-off threshold. Subvolumes that contained no protein density or did not align against the reference were removed based on a cross-correlation threshold.

Subtomograms were then extracted from 4x binned tomograms at positions determined in the 8x-binned alignments and averages were generated using the orientational parameters determined in these 8x-binned alignments. The subtomograms were aligned, progressively reducing the angular search range. During these alignments a low-pass filter at 32 Å or 35 Å was applied for the pH8 and pH6 datasets, respectively. At the end of the 4x-binned alignments a subvolume-to-subvolume distance threshold was applied to remove subvolumes that might have shifted onto identical positions on the lattice.

The remaining subvolumes were extracted from 2x-binned tomograms at their aligned positions. At this stage the datasets were split into even/odd half sets and from here on, the even/odd datasets were treated completely independently. Subvolumes with mean grey values that deviated from the dataset mean with more than ± 1 standard deviation were removed. For the even/odd datasets, independent 2x-binned references were generated by averaging their respective subvolumes using the alignment parameters determined in the 4x-binned alignments. After two more rounds of alignment the subvolumes were finally extracted from the unbinned tomograms and again independent references were generated for the respective half sets. Two more rounds of alignment in bin 1 were performed. Statistics for the sizes of the different datasets is given in S1 Fig.

The tubular structures showed a varying degree of anisotropy due to a preferential orientation of the tubes in the tomograms with respect to the tilt axis. To more appropriately handle the anisotropic distribution of subtomogram orientations, weighted averaging of the 1x binned data was performed using modified wedge masks [58] instead of a binary wedge mask. The modified wedge masks were calculated by averaging the amplitude spectra from structure-free (ice only) areas in each tomogram and therefore represent an ice/noise amplitude spectrum modulated by CTF correction, dose filtering and weighted back-projection.

The final averages were multiplied with a gaussian-filtered cylindrical mask and the resolution was determined by mask-corrected Fourier-shell correlation at the 0.143 criterion. The half maps were then averaged, sharpened (for the empirically determined B-factor values see S1 Table) and filtered to the measured resolution [59]. Visualization of tilt series, tomograms and EM-densities was performed in IMOD [51], Amira 4, UCSF Chimera [60], Coot [61] and Pymol [62].

## Atomic model building, refinement and model analysis

The resolution of our electron microscopy maps allowed us to refine existing models of the EIAV CA and to *de novo* build the last residues of CA and SP that form the CA-SP helix. At ~4 Å resolution the helical pitch is clearly visible and also large side chains (e.g. tryptophans, phenylalanines, arginines or lysines) can be identified. Small side chains and also negatively charged side chains are not clearly visible at this resolution. Therefore, rotamer refinements of residues are not possible.

Model refinements were performed into three of the EIAV GagΔMA EM maps: spheres assembled at pH6, tubes assembled at pH6, and tubes assembled at pH8. The crystal structure of EIAV CA (pdb 2EIA) [63] was used as a starting model for refinement into the map determined from spheres assembled at pH6. One CA monomer of pdb 2EIA was selected and the $CA_{NTD}$ and $CA_{CTD}$ of this monomer were independently placed into the EM-density using the rigid body fitting option in UCSF Chimera. Subsequently the linker connecting the two CA domains was joined in Coot. Additionally, the last residues of CA and the first residues of SP (T347-L359) that were not present in pdb 2EIA were manually built into our EM-density using Coot.

To account for the different monomer-monomer interactions in GagΔMA, the monomers were replicated according to the inherent 6-fold symmetry of the map, resulting in 18 symmetry related copies of GagΔMA. A map segment (defined by a mask extending 3 Angstrom around the rigid body fitted model) was extracted, and real-space coordinate refinement against the EM-density was performed using Phenix [64], which was iterated with manual model building in Coot in a similar fashion as described previously [15]. Secondary structure restraints and non-crystallographic symmetry (NCS) restraints were applied throughout all refinements.

In Phenix each iteration consisted of 5 macro cycles, in which simulated annealing was performed in every macro cycle. Atomic displacement parameter (ADP) refinement was performed at the end of each iteration.

The refined model for one EIAV GagΔMA monomer in the spherical assembly was then used as starting model for the refinement into the EM-density maps generated from tubular assemblies. The monomer was again rigid body fitted into the EM densities three times to accommodate the 3 symmetry independent copies of the EIAV GagΔMA monomer per hexamer in the tubes. As the monomers show a difference in the respective orientations of their $CA_{NTD}$ and $CA_{CTD}$, the fit was further manually refined in Coot. Subsequently, the symmetry independent monomers were expanded within the cryo-EM density to cover all potential CA-SP interactions. The refinement of these models was then done as described above for the spherical model, iteratively refining in Phenix and Coot, with the difference that no NCS

restraints were applied. The quality of all models was validated using MOLPROBITY [65] and is given in S2 Table.

The HIV-1 CA-SP1 model used for comparison was pdb 5L93 [15]. All comparisons and RMSD calculations were performed in UCSF Chimera, between the C-alpha backbone atoms selected residues. To compare all EIAV GagΔMA monomer conformations the individual monomers were first aligned on the $CA_{CTD}$ (residues 272–359). RMSD values were then calculated for the C-alpha atoms of the individual EIAV CA models including residues 143–271 for the $CA_{NTD}$ and residues 272–342 (until the end of helix 11) for the $CA_{CTD}$. Fig 4 showing the conservation of residues between HIV-1 and EIAV was performed using the 'multialign viewer' tool in UCSF Chimera.

## Data deposition

The EM-density maps and two representative tomograms have been deposited in the EMDB under accession numbers EMD-10381, EMD-10382, EMD-10383, EMD-10384, EMD-10385 and EMD-10386.

The refined models were deposited in the PDB under accession codes PDB 6T61, PDB 6T63 and PDB 6T64.

## Molecular dynamics simulations

All atom molecular dynamics simulations were employed to investigate the interactions between IP6 and EIAV's Gag lattice. In total, six atomic models of EIAV GagΔMA hexamer were built, namely, *Apo*- and *Holo*-$CASP_{143-359}$ wildtype hexamer. The starting $CASP_{143-359}$ hexamer system was refined from the Cryo-ET atomic model of the sphere EIAV GagΔMA capsid. One IP6 molecule was rigid-body docked at the binding pocket identified in Cryo-ET densities. Subsequently, 18 adjacent EIAV monomers of each central hexamer were added to complete the lattice. All models were then solvated with the TIP3P water model, neutralized by addition of NaCl, finally the total NaCl concentrations was set to 150 mM. Resulting models had total atom counts around 388 K atoms.

After model building, these systems were initially subjected to energy minimization in two stages, both using the conjugated gradient algorithm [66] with linear searching [67]. Only water and ion molecules were free to move during the first minimization stage, while the protein backbone atoms were restrained in the second stage. Convergence of the minimizations were confirmed once the variances of gradients were not greater than 1 Kcal mol$^{-1}$ Å$^{-1}$. After minimizations, the temperature of all systems were gradually increased from 50 to 310 K in 20 K increments over 5 ns. The restraints applied on backbone atoms were maintained during the thermalization stage and then gradually released from 10 to 1 Kcal mol$^{-1}$ Å$^{-2}$ over 5 ns of equilibration at 310 K and 1 atm.

Once equilibrated, production simulations of six EIAV systems were conducted on TACC Stampede 2 and NCSA Bluewaters supercomputers. *Apo*- and *Holo*-$CASP_{143-357}$ wildtype hexamers were ran for over 100 ns. Molecular dynamics simulations in this study were performed on NAMD2.12 [68] using the CHARMM36m force field [69]. During the production runs, the temperature (310 K) was maintained by employing a Langevin thermostat with a coupling factor of 0.1 ps$^{-1}$ [70]; similarly, pressure was maintained at 1 atm employing the Nosé-Hoover Langevin piston with a decay time of 1ps and a period of 2ps [71,72]. A time step of 2 fs was used for all simulations and all bonds to hydrogen atoms were constrained with the SHAKE algorithm [73]. Long-range electrostatics were calculated using the Particle-Mesh-Ewald method with a grid size of 1 Å [74] with a cutoff of 1.2 nm, as implemented in NAMD. Full

electrostatic interactions were calculated every two-time steps while nonbonded interactions were computed every time step.

## Supporting information

**S1 Fig. EIAV and HIV-1 Gag and CA sequence similarity. (A)** Schematic representation of HIV-1 and EIAV Gag, and the truncated Gag variants used in this study. The cleavage sites are annotated by arrows and the first residue of each individual domain is shown. The $CA_{NTD}$ and $CA_{CTD}$ are colored cyan and orange, respectively. The SP region is colored red. Abbreviated protein names in parenthesis. **(B)** Sequence alignment between HIV-1 CA-SP1 and EIAV CA-SP. The background of the sequence is colored as in (A) to indicate the location of the $CA_{NTD}$, $CA_{CTD}$ or SP. Secondary structure assignments (alpha-helices) are indicated with dashed lines. Above the sequence alignment the conservation of the respective residues and existing charge variations are shown, in gray and blue/red respectively; a blue bar indicates a positive charge variation from EIAV to HIV-1, conversely a red bar indicates a negative charge variation from EIAV to HIV-1. Conserved lysine residues are in indicated by dashed rectangles. **(C)** Molecular representations using the licorice representation of IP6 and IP5. It is clearly shown that IP5 lacks the axial phosphate present in IP6.
(TIF)

**S2 Fig. Screening examples of EIAV Gag and Gag truncation assembly.** (A,B) Representative low and high magnification images of Gag and GagΔMA assembled in the absence (red) or presence (blue) of 10 μM IP6 at pH 8. The number of VLPs (spheres-purple, tubes-green) per 55μm$^2$ for no fewer than five representative images for each condition. Center lines show the medians; box limits indicate the 25th and 75th percentiles as determined by R software; whiskers extend to minimum and maximum values; data points are plotted as circles. (C) Representative low and high magnification images of native GagΔMA assembled in the absence (red) and presence (blue) of 10 μM IP6 at pH 6. Tubes assembled in the absence of IP6 were the same, by negative stain EM, as tubes assembled with GagΔMAΔp9 containing an ectopic serine at the N-terminus. In the presence of IP6, protein formed multi-layered, spherical VLPs. (D) Representative low and high magnification images of GagΔMAΔNC$^{6-76}$ assembled in the absence (red) or presence (blue) of IP6 at pH 6. Very few VLPs were observed; see purple triangle, compared to the amount observed for HIV-1 [17]. The mean value of counted particles is given in italics in the bar charts.
(TIF)

**S3 Fig. Effect of IP6 on Infectious particle production. (A)** The effect of IPPK KO on the production of infectious EIAV and HIV virus particles. Relative to WT, bars represent the average and whiskers the standard deviation of no fewer than three replicates. **(B)** Western blots were performed on cell lysates and released virus from HEK293T WT and IPPK KO cells for EIAV and HIV. EIAV was probed with RbαEIAV-CANC and HIV with MsαHIVp24.
(TIF)

**S4 Fig. Cryo-ET and subtomogram averaging of EIAV GagΔMA. (A)** Radial orthoslices through the final sharpened map from spheres assembled at pH6, filtered to 8 Å resolution. Protein density is white. The level of the $CA_{NTD}$, $CA_{CTD}$ and SP is indicated with dashed boxes. **(B)** Fourier shell correlations (FSC) between independent halfsets for EIAV GagΔMA spheres (green) and tubes (orange) assembled at pH6, and spheres (pink) and tubes (blue) assembled at pH8. The resolution measured at the 0.143 FSC criterion is indicated with lines. The Nyquist frequency for the dataset of particles assembled at pH8, which was acquired with

a pixel size of 1.35 Å is indicated at 2.7 Å.
(TIF)

**S5 Fig. A positively charged region at the base of EIAV CA$_{CTD}$ potentially interacts with nucleic acids. A)** Isosurface representations of EIAV CA-SP from spherical and tubular assemblies at pH8 and pH6 as seen from the inside of the particle. All structures have been filtered to 8 Angstrom resolution. The additional densities in the spheres and tubes assembled at pH6 are highlighted in yellow. The corresponding positions in the sphere structure assembled at pH8 are circled in yellow. The red dashed line indicates the tube axis, showing that the additional densities are absent along the direction of the tube, where no curvature between the CA$_{CTD}$ dimer is found. This observation of an additional density being recruited to the base of the CA$_{CTD}$ by basic residues is reminiscent of a similar interface observed in M-PMV [75]. **B)** The three positively charged residues (green) in the linker connecting helix 10 and helix 11 are positioned to interact with the additional density (displayed as an isosurface in yellow). **C)** (Top) Low and high magnification negative staining TEM images of GagΔMA (RHR:AAA) VLPs assembled at pH 6 without (red) and with (blue) 10 μM IP6. (Bottom) Quantification of the number of VLPs counted for no fewer than five representative TEM images at pH 6 without and with 10 μM IP6. The mean value of counted particles is given in italics in the bar charts.
(TIF)

**S6 Fig. Stability of the 6HB is independent of the presence of IP6. A)** Structural changes observed following ~150 ns of MD simulations of CA$_{CTD}$SP without and with bound IP6 for model 1 (correct model) and model 2 (model with registry shift in the SP helix). We ran one simulation with an incorrect model with a n+1 registry shift of the SP helix, clearly showing the loss of 6HB bundle stability and partial unfolding. **B)** Root mean squared deviations (RMSDs) of the central hexamer during simulations. **C)** Root mean squared fluctuations were averaged over six central monomers for model 1 with standard deviations shown for each residue. RMSFs are good indicators of protein flexibility.
(TIF)

**S7 Fig. Q350A mutation reduces overall particle number but not sensitivity to IP6.** (A) Representative low and high magnification negative stain EM images of *in vitro* assembled GagΔMA (Q350A) in the absence (red) and presence (blue) of 10 μM IP6. Compared to GagΔMA (Fig 1), the number of VLPs formed in the absence and presence of IP6 is much lower. However, the stimulatory effect of IP6 is still apparent. (B) Relative infectious particle production of wild type EIAV Gag (WT) and Gag (Q350A) point mutation VSV-G-pseudo-typed provirus in 293FT cells. Graphs show the average and standard deviation of three independent experiments; dots show individual data points. The mean value of counted particles is given in italics in the bar charts.
(TIF)

**S8 Fig. Diameter of different lentivirus VLPs.** In vitro assembled VLPs were imaged via negative stain EM. From the latter the diameters of the particles were determined.
(TIF)

**S1 Table. Cryo-ET data acquisition and subtomogram averaging statistics.**
(PDF)

**S2 Table. Model refinement statistics.**
(PDF)

**S1 Movie. Immature EIAV CA-SP assembly in GagΔMA spheres assembled at pH6.** A 3D visualization of the 3.7 Å EIAV CA-SP structure **i**n GagΔMA spheres assembled at pH6. The movie displays both tubes and spheres at pH6, the structure and model are derived from spheres. (MOV)

**S2 Movie. A guided tour comparing the structural differences of EIAV CA-SP assemblies in tubes and spheres at pH6 and pH8.** A tour comparing key interfaces and variable interactions of EIAV CA-SP described in Fig 3. (MOV)

**S3 Movie. Comparison of immature CASP interfaces in EIAV and HIV-1.** A comparison of the CASP assembly in EIAV and HIV-1, as described in Fig 4. (MOV)

**S4 Movie. EIAV six-helix bundle stability without and with IP6.** All-atom molecular dynamics simulation of EIAV CA$_{CTD}$-SP model 1 as described in S6 Fig. A moving average of the 3D coordinates with a window size of 1 ns is employed to remove high frequency motions from the movie. IP6 is shown in licorice representation. (MOV)

## Acknowledgments

We thank Wim Hagen (EMBL Heidelberg), the IST Austria scientific computing service unit, Scientific Computing at MRC-LMB, as well as the EMBL IT support unit for assistance.

## Author Contributions

**Conceptualization:** Robert A. Dick, Volker M. Vogt, John A. G. Briggs, Florian K. M. Schur.

**Data curation:** Robert A. Dick, Marc C. Johnson, Volker M. Vogt, Juan R. Perilla, John A. G. Briggs, Florian K. M. Schur.

**Formal analysis:** Robert A. Dick, Chaoyi Xu, Dustin R. Morado, Vladyslav Kravchuk, Florian K. M. Schur.

**Funding acquisition:** Marc C. Johnson, Volker M. Vogt, Juan R. Perilla, John A. G. Briggs, Florian K. M. Schur.

**Investigation:** Robert A. Dick, Chaoyi Xu, Vladyslav Kravchuk, Clifton L. Ricana, Terri D. Lyddon, Arianna M. Broad, J. Ryan Feathers, Florian K. M. Schur.

**Methodology:** Robert A. Dick, Dustin R. Morado, Juan R. Perilla, Florian K. M. Schur.

**Project administration:** Robert A. Dick, Volker M. Vogt, John A. G. Briggs, Florian K. M. Schur.

**Resources:** Robert A. Dick, Marc C. Johnson, Volker M. Vogt, Juan R. Perilla, John A. G. Briggs, Florian K. M. Schur.

**Supervision:** Robert A. Dick, Marc C. Johnson, Volker M. Vogt, Juan R. Perilla, John A. G. Briggs, Florian K. M. Schur.

**Validation:** Robert A. Dick, Marc C. Johnson, Volker M. Vogt, Juan R. Perilla, John A. G. Briggs, Florian K. M. Schur.

**Visualization:** Robert A. Dick, Chaoyi Xu, Florian K. M. Schur.

**Writing – original draft:** Robert A. Dick, Volker M. Vogt, John A. G. Briggs, Florian K. M. Schur.

**Writing – review & editing:** Robert A. Dick, Chaoyi Xu, Vladyslav Kravchuk, Marc C. Johnson, Volker M. Vogt, Juan R. Perilla, John A. G. Briggs, Florian K. M. Schur.

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
