## [Decision Letter · Decision Letter 0]

11 Sep 2019

Dear Dr. Schur,

Thank you very much for submitting your manuscript "Structures of immature EIAV Gag lattices reveal a conserved role for IP6 in lentivirus assembly" (PPATHOGENS-D-19-01294) for review by PLOS Pathogens. We apologize for the delay in getting back to you, but in the summer period it was difficult to secure reviewers. The reviewers found that your study provides interesting insight into the assembly of lentivirus particles, and as such they are an important contribution to the field. As you will see, the main concern was expressed by reviewer 1, and has to do with the way particle release is measured. We feel that it is indeed important for the conclusions you reach to have a way of distinguishing between particles that are correctly assembled but have a maturation default, from released particles that are non-infectious because of an aberrant assembly, and therefore only measuring the fraction of released infectious particles is not sufficient to support your conclusions. We therefore expect that you will be able to answer this point appropriately in the revised version. You will also see that reviewer 2 requests an additional control to validate the observed IP6 density, which if you can provide would be nice, but we do not find it essential for the validity of the paper. Of course, we cannot promise publication until we have seen your revised manuscript.

(1) A letter containing a detailed list of your responses to the review comments and a description of the changes you have made in the manuscript. Please note while forming your response, if your article is accepted, you may have the opportunity to make the peer review history publicly available. The record will include editor decision letters (with reviews) and your responses to reviewer comments. If eligible, we will contact you to opt in or out.

(2) Two versions of the manuscript: one with either highlights or tracked changes denoting where the text has been changed; the other a clean version (uploaded as the manuscript file).

Additionally, to enhance the reproducibility of your results, PLOS recommends that you deposit your laboratory protocols in protocols.io, where a protocol can be assigned its own identifier (DOI) such that it can be cited independently in the future. For instructions see http://journals.plos.org/plospathogens/s/submission-guidelines#loc-materials-and-methods

We hope to receive your revised manuscript within 60 days. If you anticipate any delay in its return, we ask that you let us know the expected resubmission date by replying to this email. Revised manuscripts received beyond 60 days may require evaluation and peer review similar to that applied to newly submitted manuscripts.

[LINK]

Sincerely,

Félix A. Rey

Associate Editor

PLOS Pathogens

Michael Malim

Section Editor

PLOS Pathogens

Kasturi Haldar

Editor-in-Chief

PLOS Pathogens

orcid.org/0000-0001-5065-158X

Grant McFadden

Editor-in-Chief

PLOS Pathogens

orcid.org/0000-0002-2556-3526

Reviewer's Responses to Questions

**Part I - Summary**

Reviewer #1: In this study, the authors examine the role of IP6 in the assembly of equine infectious anemia virus (EIAV) and several other lentiviruses. By using a variety of approaches, they observe that IP6 markedly stimulates the assembly of EIAV and the other lentiviruses. EIAV Gag is observed to assemble in vitro into several morphologically distinct tubes and sphere; interestingly, the Gag lattice in all these structures is in the immature conformation. Both EIAV and HIV-1 maintain similar immature inferfaces, and both lentiviral Gag proteins bind IP6 via conserved Lys residues in the CA C-terminal domain and adjacent spacer peptide.

In general, the study provides some interesting insights into lentiviral assembly and the role of IP6 in assembly. Some of the results might have been predicted based on earlier work with HIV-1, whereas other findings were unexpected. One major issue, related to how particle production is measured in cell-based assays, needs to be addressed. In their previous work, Dick and colleagues (Nature 2018) measured virus particle production by quantifying the amount of infectious virus released into the supernatant. This approach has a serious limitation: it does not distinguish between effects on particle assembly/release and effects on particle maturation/infectivity. Thus it is not clear from the previous work whether KO of IPPK, a key enzyme in the IP6 biosynthetic pathway, impairs particle assembly/release, maturation/infectivity, or both. This information is vital to understanding the role of IP6 in lentivirus replication. In the current study, the authors use the same approach to evaluate the effect of IPPK KO on virus particle production.

Reviewer #2: In this manuscript, Dick et al. characterise the immature Gag lattice of the lentivirus EIAV. They first recombinantly express different Gag constructs and establish the conditions for their assembly. They do this in the presence or absence of IP6, and find that, while IP6 promotes EIAV Gag assembly, the dependence of EIAV on IP6 is lower than that for HIV. They then structurally characterise some of the assembled lattices by cryo-electron tomography and subtomogram averaging, which allows the authors to produce atomic models for different rearrangements of the capsid domains. They use these atomic models for: (1) studying the key interactions to maintain the lattice; (2) comparing the capsid proteins of EIAV and HIV; and (3) producing mutants to confirm some of their findings. Finally, the authors show that other lentiviruses also require IP6.

Overall, this is a very complete piece of work, with a state-of-the-art subtomogram averaging results, complemented by mutagenesis experiments that confirm the authors conclusions.

Reviewer #3: This paper studied the role of IP6 in the immature particle assembly of EIAV through in vitro and in viro assembly assays, cryo-ET reconstruction and molecular modeling methods. The major results and conclusions of the papers are: (1) IP6 promotes in vitro assembly of EIAV Gag proteins into immature particle structures via conserved lysine residues within the CA-CTD and SP. (2) This function of IP6 is conserved for all lentivirus Gag proteins, although at varied magnitudes. (3) EIAV Gag lattice was resolved from in vitro assembled tubular and spherical structures, at sub-4Å resolution using cryo-ET and subtomogram averaging techniques. (4) When comparing that with HIV-1, the molecular contacts involved with CA-CTD organization are conserved, however the CA-NTD arrangement are not conserved.

Comments, concerns or questions:

1) Although stimulated by IP6, in vitro assembly of EIAV Gag can happen without presence of IP6 (line 331-335). The authors think this is because that in EIAV most of the interactions of the six-helix bundle are hydrophobic in nature, which strengthen the six-helix bundle structure. In contrast to HIV-1, “MD simulations the EIAV 6HB was stable both with and without IP6 likely due to additional side chain contacts between helices in the six-helix bundle” (line 404-405). Can you perform a sequence alignment of this region of all lentiviruses studied in the paper and address if this property is unique for EIAV?

2) Line 264-269: EIAV CA was found to have a shorter helix 1, and therefore in EIAV Gag lattice, the putative dimer interface between neighboring Gag hexamers (found in HIV-1 lattice) is missing. According to HIV-1 mature CA lattice, helix 1 is important in intra-hexamer interactions. Could this be one of the reasons that in the in vitro assembly systems, EIAV Gag always form immature lattice, not mature lattice? Is this property unique for EIAV?

3) The authors found that the EIAV CA-NTD organization is not the same as that in HIV-1. Several molecular contacts in HIV-1 Gag lattice are missing in EIAV which include: (a) there is no intra-hexamer interactions among CA-NTD (line 238-239) (b) there is no dimeric inter-hexamer interaction formed by helix 1 (c) there is no strong contact between CA-NTD and CA-CTD (line 263-264). The studied EIAV assemblies are rather small: narrow tube 35nm, wide tube 70nm and spherical particles are 100nm. In general, retroviruses have a larger dimension. Is it possible that these missing contacts actually present in the authentic particles?

4) Line 109-110: The authors suggests that “the retroviral CA-NTD and CA-CTD are independent structural entities that act autonomously in regulating immature virus particle diameter and curvature”. This idea was repeated at line 363-364: “The spatial separation of EIAV CA-NTD and CA-CTD, and the lack of strong interactions between them, allow these two halves of CA to behave almost as independent entities, as suggested previously for other retroviruses”. This statement seems overly strong. In HIV-1, helix 11 and helix 8 interact with MHR and helix 7. This interaction would limit the orientation of NTD relative to CTD and the role of NTD in deciding the curvature of the Gag lattice.

5) Line 292-295: Line 292 lists the SP sequence as (LAKAL) in which the residue 359 is L. However in line 295, the last residue is listed as T395. Please clarify.

6) Supplement figure 4, the electron density looks low resolution. A better figure is desired.

**Part II – Major Issues: Key Experiments Required for Acceptance**

Reviewer #1: 1. Supp Fig. 3. The authors measure the effect of IPPK KO on EIAV and HIV-1 particle production by measuring the infectivity of virus produced from KO cells. As mentioned above, this approach does not differentiate between effects on particle production and effects on infectivity. The authors need to perform virus assembly assays in which levels of HIV-1 and EIAV Gag in the cell and virus fractions are quantified so that virus release efficiency can be determined, a infectivity values need to be normalized based on viral input (measured by RT or capsid protein).

2. P. 16. The authors speculate about the role of IP6, or potentially other positively charged molecules, in EIAV assembly in vivo. They suggest that, in the cells used, EIAV infectivity is significantly lower than that of HIV-1. Again, this issue needs to be addressed by measuring the efficiency of virus particle production and determining the relative infectivity (normalized for levels of RT or CA) of the EIAV stocks. These crucial experiments will resolve this issue.

3. Line 425. The authors discuss how future work will evaluate the effect of IP6 depletion on assembly, budding and infectivity. This work will need to be performed by measuring virus release efficiency and specific particle infectivity as described above.

4. Supp Fig. 7. Have the authors measured the efficiency of virus particle production of the Q350A mutant in IPPK KO cells? This result would be interesting.

Reviewer #2: My only major concern is related to the assignment of the density for IP6. In order to be certain that this density corresponds indeed to IP6, the authors should obtain an immature lattice average without it (i.e. a negative control), for example using the EIAV GAGdeltaMA preparation with 0µM IP. This is specially required as IP6 sits at the symmetry axis, and non-relevant densities can appear there due to symmetrisation.

Reviewer #3: Comments, concerns or questions:

1) Although stimulated by IP6, in vitro assembly of EIAV Gag can happen without presence of IP6 (line 331-335). The authors think this is because that in EIAV most of the interactions of the six-helix bundle are hydrophobic in nature, which strengthen the six-helix bundle structure. In contrast to HIV-1, “MD simulations the EIAV 6HB was stable both with and without IP6 likely due to additional side chain contacts between helices in the six-helix bundle” (line 404-405). Can you perform a sequence alignment of this region of all lentiviruses studied in the paper and address if this property is unique for EIAV?

**Part III – Minor Issues: Editorial and Data Presentation Modifications**

Reviewer #1: Minor: Fig. 6A. The sequence of SP1 is incorrect, as is the placement of the CA-SP1 cleavage site. The correct sequence is KARVL/AEAMSQ, where / denotes the cleavage site.

Reviewer #2: - Can the authors estimate the occupancy of IP6 in their averages?

- It would be good if the authors could provide a list/table of the conditions attempted to produce a mature EIAV lattice. Related to this, there is a reference missing in line 473

Reviewer #3: Comments, concerns or questions:

2) Line 264-269: EIAV CA was found to have a shorter helix 1, and therefore in EIAV Gag lattice, the putative dimer interface between neighboring Gag hexamers (found in HIV-1 lattice) is missing. According to HIV-1 mature CA lattice, helix 1 is important in intra-hexamer interactions. Could this be one of the reasons that in the in vitro assembly systems, EIAV Gag always form immature lattice, not mature lattice? Is this property unique for EIAV?

3) The authors found that the EIAV CA-NTD organization is not the same as that in HIV-1. Several molecular contacts in HIV-1 Gag lattice are missing in EIAV which include: (a) there is no intra-hexamer interactions among CA-NTD (line 238-239) (b) there is no dimeric inter-hexamer interaction formed by helix 1 (c) there is no strong contact between CA-NTD and CA-CTD (line 263-264). The studied EIAV assemblies are rather small: narrow tube 35nm, wide tube 70nm and spherical particles are 100nm. In general, retroviruses have a larger dimension. Is it possible that these missing contacts actually present in the authentic particles?

4) Line 109-110: The authors suggests that “the retroviral CA-NTD and CA-CTD are independent structural entities that act autonomously in regulating immature virus particle diameter and curvature”. This idea was repeated at line 363-364: “The spatial separation of EIAV CA-NTD and CA-CTD, and the lack of strong interactions between them, allow these two halves of CA to behave almost as independent entities, as suggested previously for other retroviruses”. This statement seems overly strong. In HIV-1, helix 11 and helix 8 interact with MHR and helix 7. This interaction would limit the orientation of NTD relative to CTD and the role of NTD in deciding the curvature of the Gag lattice.

5) Line 292-295: Line 292 lists the SP sequence as (LAKAL) in which the residue 359 is L. However in line 295, the last residue is listed as T395. Please clarify.

6) Supplement figure 4, the electron density looks low resolution. A better figure is desired.

PLOS authors have the option to publish the peer review history of their article (what does this mean?). If published, this will include your full peer review and any attached files.

Reviewer #1: No

Reviewer #2: No

Reviewer #3: No

---

## [Editor Report · Decision Letter 1]

26 Nov 2019

Dear Dr. Schur:

Thank you very much for submitting your revised manuscript "Structures of immature EIAV Gag lattices reveal a conserved role for IP6 in lentivirus assembly" (PPATHOGENS-D-19-01294R1) for review by PLOS Pathogens. Your manuscript was fully evaluated at the editorial level, and was found to be improved with respect to the original submission. There are two minor issues that we would like you to consider before formal acceptance, as you will see listed below. In addition, when you are ready to resubmit, please be prepared to provide the following:

(1) A letter containing a detailed list of your responses to the review comments and a description of the changes you have made in the manuscript. Please note while forming your response, if your article is accepted, you may have the opportunity to make the peer review history publicly available. The record will include editor decision letters (with reviews) and your responses to reviewer comments. If eligible, we will contact you to opt in or out.

(2) Two versions of the manuscript: one with either highlights or tracked changes denoting where the text has been changed; the other a clean version (uploaded as the manuscript file).

We hope to receive your revised manuscript within 60 days or less. If you anticipate any delay in its return, we ask that you let us know the expected resubmission date by replying to this email.

[LINK]

Sincerely,

Félix A. Rey

Associate Editor

PLOS Pathogens

Michael Malim

Section Editor

PLOS Pathogens

Kasturi Haldar

Editor-in-Chief

PLOS Pathogens

orcid.org/0000-0001-5065-158X

Grant McFadden

Editor-in-Chief

PLOS Pathogens

orcid.org/0000-0002-2556-3526

The revised manuscript by Dick et al is significantly improved with respect to the original version as the authors have now quantified the amount of released EIAV particles from wild type and IPPK KO cells, showing that there is only a minor decrease in particle release which correlates with the drop in infectiousness, contrary to HIV-1 infected cells for which there is apparently no particle released into the supernatant in the IPPK KO cells. This was one of the main issues raised by reviewer 1 of the original version. The authors have also adequately answered the other questions, except for the request by reviewer 2, who asked to provide a control reconstruction of immature assemblies in the absence of IP6. We had editorially considered that answering to such a request was not a requirement for acceptance, given the amount of work involved and that the observed density for IP6 conformed to the expected shape for this molecule, that it was as strong as the surrounding protein density (including in regions where only 2-fold averaging instead of 6-fold was done). The new version is now acceptable for publication, provided that the authors pay attention to two minor issues:

1. In the discussion, I suggest that the authors add a sentence considering a potential basal immature assembly in the absence of IP6, which would be similar for all lentiviruses but that is detected here in the case of EIAV because the cells used are poorly infectable by this virus. As the same cells are much more susceptible to HIV1 infection, the difference in particle release in the case of EIAV is not very different to basal, whereas it would be more than two logs in the case of HIV1. Depending on the infectivity assays and on the reactivity of the antibodies used in the western blot and on the, this basal assembly in the case of HIV-1 could be missed. In other words, it could be that the striking difference in the requirement for IP6 suggested by Fig. S3 is due to the cell line used for the experiment and does not reflect an intrinsic difference between the two viruses.

2. In Figure 1, I suggest that the authors take a look at the “representative images” that they provide in relation with the quantification provided. For instance, panel A suggests that there are predominantly spheres at 0uM IP and at 10 uM IP5, whereas the quantification in panel B indicates that there are mainly tubes under these conditions. Similarly, panel C (or at least, the enlargement shown) suggests that there are essentially tubes at 10 uM IP5, whereas the quantification in panel D indicates more spheres than tubes. This can be confusing to some readers, so if possible, please provide more representative micrographs in each case (or at least, an enlargement that matches the quantification shown immediately below the micrograph)

---

## [Editor Report · Decision Letter 2]

11 Dec 2019

Dear Dr. Schur,

We are pleased to inform that your manuscript, "Structures of immature EIAV Gag lattices reveal a conserved role for IP6 in lentivirus assembly", has been editorially accepted for publication at PLOS Pathogens. 

Before your manuscript can be formally accepted and sent to production, you will need to complete our formatting changes, which you will receive by email within a week. Please note that your manuscript will not be scheduled for publication until you have made the required changes.

IMPORTANT NOTES

(1) Please note, once your paper is accepted, an uncorrected proof of your manuscript will be published online ahead of the final version, unless you’ve already opted out via the online submission form. If, for any reason, you do not want an earlier version of your manuscript published online or are unsure if you have already indicated as such, please let the journal staff know immediately at plospathogens@plos.org.

(2) Copyediting and Proofreading: The corresponding author will receive a typeset proof for review, to ensure errors have not been introduced during production. Please review the PDF proof of your manuscript carefully, as this is the last chance to correct any errors. Please note that major changes, or those which affect the scientific understanding of the work, will likely cause delays to the publication date of your manuscript. 

(3) Appropriate Figure Files: Please remove all name and figure # text from your figure files. Please also take this time to check that your figures are of high resolution, which will improve the readbility of your figures and help expedite your manuscript's publication. Please note that figures must have been originally created at 300dpi or higher. Do not manually increase the resolution of your files. For instructions on how to properly obtain high quality images, please review our Figure Guidelines, with examples at: http://journals.plos.org/plospathogens/s/figures.

(4) Striking Image: Please upload a striking still image to accompany your article if one is available (you can include a new image or an existing one from within your manuscript). Should your paper be accepted, this image will be considered for our monthly issue image and may also appear on our website to feature your article. Please upload this as a separate file, selecting "striking image" as the file type upon upload. Please also include a separate "Other" file with a caption, including credits and any potential copyright information. Please do not include the caption in the main article file. If your image is from someone other than yourself, please ensure that the artist has read and agreed to the terms and conditions of the Creative Commons Attribution License at http://journals.plos.org/plospathogens/s/content-license. Please note that PLOS cannot publish copyrighted images.

(5) Press Release or Related Media: If your institution or institutions have a press office, please notify them about your upcoming paper at this point, to enable them to help maximize its impact. If they will be preparing press materials for this manuscript, please inform our press team in advance at plospathogens@plos.org as soon as possible. We ask that you contact us within one week to plan ahead of our fast Production schedule. If you need to know your paper's publication date for related media purposes, you must coordinate with our press team, and your manuscript will remain under a strict press embargo until the publication date and time. This means an early version of your manuscript will not be published ahead of your final version. 

(6)  PLOS requires an ORCID iD for all corresponding authors on papers submitted after December 6th, 2016. Please ensure that you have an ORCID iD and that it is validated in Editorial Manager.  To do this, go to ‘Update my Information’ (in the upper left-hand corner of the main menu), and click on the Fetch/Validate link next to the ORCID field.  This will take you to the ORCID site and allow you to create a new iD or authenticate a pre-existing iD in Editorial Manager

(7) Update your Profile Information: Now that your manuscript has been provisionally accepted, please log into Editorial Manager and update your profile, if needed. Go to https://www.editorialmanager.com/ppathogens, log in, and click on the "Update My Information" link at the top of the page. Please update your user information to ensure an efficient production and billing process. 

(8) LaTeX users only: Our staff will ask you to upload a TEX file in addition to the PDF before the paper can be sent to typesetting, so please carefully review our Latex Guidelines http://journals.plos.org/plospathogens/s/latex in the meantime.

(9) If you have associated protocols in protocols.io, please ensure that you make them public before publication to guarantee immediate access to the methodological details.

Best regards,

Félix A. Rey

Associate Editor

PLOS Pathogens

Michael Malim

Section Editor

PLOS Pathogens

Kasturi Haldar

Editor-in-Chief

PLOS Pathogens

orcid.org/0000-0001-5065-158X

Grant McFadden

Editor-in-Chief

PLOS Pathogens

orcid.org/0000-0002-2556-3526

The authors have adequately addressed the minor editorial suggestions made in the previous round of revision, and the manuscript is now acceptable for publication. This is a very high quality paper with data obtained using state-of-the-art electron cryo-tomography and subtomogram averaging methodologies that the authors have strongly contributed to develop. It will certainly be of interest the broad readership of PLoS Pathogens.
---

## [Editor Report · Acceptance letter]

16 Jan 2020

Dear Dr. Schur,

We are delighted to inform you that your manuscript, "Structures of immature EIAV Gag lattices reveal a conserved role for IP6 in lentivirus assembly," has been formally accepted for publication in PLOS Pathogens.

Best regards,

Kasturi Haldar

Editor-in-Chief

PLOS Pathogens

orcid.org/0000-0001-5065-158X

Michael Malim

Editor-in-Chief

PLOS Pathogens

orcid.org/0000-0002-7699-2064